# Human placental mesenchymal stem cells ameliorates premature ovarian insufficiency via modulating gut microbiota and suppressing the inflammation in rats

Shudan Liu[1,2], Ting Wang[1], Yuanyuan Liu[1], Yiwei Li[1], Junbai Ma[1], Qikuan Hu[1]*, Hao Wang ![ORCID][1]*, Xiaoxia Zhang[3]*

**1** School of Basic Medical Sciences, Ningxia Medical University, Yinchuan, Ningxia, China, **2** Ningxia Medical University General Hospital, Yinchuan, Ningxia, China, **3** College of Traditional Chinese Medicine, Ningxia Medical University, Yinchuan, Ningxia, China

\* huqikuan@163.com (QH); wanghaograduate@126.com (HW); zxx1216@163.com (XZ)

## Abstract

The core objective of this study was to explore the effects and potential mechanism of human placental mesenchymal stem cells (PMSCs) in improving early-onset ovarian dysfunction (POI). Mesenchymal stem cells with multidirectional differentiation ability were isolated from human placenta tissue and a culture system of human PMSCs was constructed for this study. Subsequently, we successfully constructed POI rat models using cisplatin induction. We randomly divided these models into four groups: CON group (blank control), MOD group (POI model), MED group (hormone therapy), and PMSC group (PMSCs therapy). Then, we compared the differences in estrus cycle, ovarian index, ovarian weight, and ovarian histopathological features, as well as hormones and inflammatory factors in rats of diverse groups. The content of lipopolysaccharide (LPS) in plasma was determined by limulus reagent kit. 16S rDNA sequencing was used to evaluate the changes in gut flora composition. Further, we investigated short-chain fatty acids (SCFAs) in the metabolites of rat gut microbiota by gas chromatography-mass spectrometry (GC-MS). As a result, we successfully established an efficient cell culture system of human placental mesenchymal stem cells (PMSCs) *in vitro*. Then, we evaluated the effects of PMSC intervention on POI rats: Compared to the untreated MOD group, the estrous cycle of rats in the PMSC group gradually became regular, the ovarian weight and ovarian index were significantly increased, and the ovarian tissue structure was improved by showing an increase in the number of follicles and a decrease in the number of atretic follicles. Moreover, PMSC intervention significantly affected plasma sex hormones by a major impact on follicle stimulating hormone (FSH) and Estradiol (E2). In terms of inflammatory factors, PMSC intervention decreased the levels of proinflammatory cytokines including interleukin (IL)-1β, IL-33, IL-6, and tumor necrosis factor (TNF)-α in plasma and ovarian tissue of POI rats. Meanwhile, the expression of anti-inflammatory cytokine IL-10 was increased. In addition, we found that there was ectopic lipopolysaccharide (LPS) in the blood of POI rats, which could be significantly reduced by PMSC intervention. Intestinal microbiota sequencing and analysis showed that after PMSC intervention, the phyla abundances of

**Data availability statement:** All relevant data are within the manuscript and its Supporting Information files.

**Funding:** This study was supported by the Key Research and Development Program of Ningxia, China (Grant No. 2023BEG02011); the Ningxia Gut Homeostasis and Chronic Disease Prevention and Treatment Scientific and Technological Innovation Team, China (Grant No. 2022BSB03112); the Leading Talents in Science and Technology in Ningxia Hui Autonomous Region (Grant No. 2023GKLRLX17).

**Competing interests:** The authors have declared that no competing interests exist.

*Firmicutes*, *Bacteroides*, and *Proteobacteria* showed remarkable differences between the MOD and PMSC groups (P < 0.05). Further genus analysis showed that PMSC treatment had a major influence in gut microbitoa by increasing the abundances of *Turicibacter* and *Desulfovibrio*, as well as reducing *Alloprevotella*, *Parabacteroides*, *Rikenellaceae_RC9_ gut_group*, and *Rikenella*. The changes of SCFAs in intestinal microbial metabolites of rats after PMSC intervention were analyzed: caproic acid level was markedly increased, butyric acid showed a decreased trend. Notably, we found a closed and complicated potential correlation among differential microbiota, inflammatory factors and hormones after PMSCs intervention. Collectively, this study have successfully established a suitable cultured PMSCs that can effectively promote the improvement of reproductive function in POI rats and achieve therapeutic effects by regulating the inflammatory response and reshaping the gut intestinal microbiota.

## Introduction

Premature ovarian insufficiency (POI) characterized by a decline in ovarian function in women before the age of 40, represents a gynecological, endocrine disease with an increasing incidence in recent years. According to a meta-analysis of the global prevalence of POI, the incidence of the disease is about 3.7%, and the trend is significantly younger [1]. POI is a significant health challenge for women, which not only has a medical impact but also has a profound impact on women's mental and reproductive health. In most cases, premature ovarian failure or follicle depletion is caused by a combination of complex factors. Although genetic, autoimmune, and iatrogenic factors are known to be involved, the specific cause of most cases remains unclear [2]. POI can lead to irregular menstruation, infertility, and a range of health problems, with the central cause being a lack of estrogen, which affects women throughout their lifetime [2]. In the long term, POI is strongly associated with health problems such as increased risks of cardiovascular disease, osteoporosis, and cognitive decline. More seriously, studies have shown that POI is also associated with an increased risk of early death [3], further highlighting the serious threat POI poses to women's health. The main characteristics of POI are insufficient ovarian sex hormone secretion and the decline of ovarian reserve function. These two conditions work together to accelerate the decline of ovarian function, leading to early menopause in women [4].

With the rapid development of regenerative medicine, stem cell therapy has become a promising therapeutic strategy for repairing ovarian tissue damage [5,6]. In particular, mesenchymal stem cells (PMSCs) from the placenta have achieved remarkable results in treating early-onset ovarian insufficiency (POI), bringing a new therapeutic dawn for POI patients. The study confirmed that PMSC transplantation effectively restores ovarian function in mice with cyclophosphamide-induced premature ovarian failure [7]. As a precious biological resource, the human placenta is a feasible source of mesenchymal stem cells. PMSCs hold great potential for treating POI due to their multiple properties, including their excellent multidirectional differentiation ability, immunomodulatory function, and the ability to promote tissue repair and regeneration [8].

There is growing evidence that POI is often strongly associated with elevated systemic and local inflammatory markers. In women with POI, significantly increased levels of pro-inflammatory cytokines such as IL-1β, IL-6, and TNF-α were observed [9]. The inflammatory process leads to ovarian damage in several ways, including activation of the immune system, which may trigger an autoimmune response against the ovarian tissue, leading to premature depletion of the follicular pool. In addition, autoimmune disease including autoimmune

ovaritis, as well as a range of systemic diseases, such as autoimmune thyroid disease and Addison's disease, have been strongly associated with POI [10]. It has been reported that human placental mesenchymal stem cell exosomes (hPMSCs-Exos) can increase the level of anti-inflammatory factor IL-10 while reducing the level of other inflammatory factors such as TNF-α, IL-8, IL-1β, and NF-κB, and effectively reduce the inflammatory response [9]. These changes effectively inhibit follicle apoptosis, a critical factor in the self-damage of ovarian structure [11]. This discovery provides a new theoretical basis and therapeutic direction, potentially leading to more effective treatments for POI using PMSCs.

The link between female reproductive health and gut microbiota has received increasing attention [12]. Studies have gradually revealed the close relationship between intestinal flora and early-onset ovarian insufficiency (POI), and the imbalance of flora may participate in and affect the ovarian injury process caused by chemotherapy drugs and then promote the occurrence of POI. The secretion and production of metabolites of intestinal flora is an essential mechanism for their participation in host physiological and pathological processes. Among them, metabolites such as short-chain fatty acids (SCFAs) can play a role in the intestinal tract and enter the blood circulation through the intestinal barrier, further affecting the function of multiple external organs [13]. Current studies have found that various metabolites are closely related to host health, including bile acids, SCFAs, methylamine, polyphenols, and indole [14–16]. In particular, SCFAs produced by the gut microbiota have been shown to regulate intestinal homeostasis, lipid metabolism, insulin resistance, and inflammatory responses. Therefore, adjusting the dysregulation of intestinal flora in patients with POI may help reduce the chronic inflammatory response [17,18] and provide new strategies and approaches for treating POI, potentially leading to significant advancements in the field.

Our research thoroughly investigated the effect of PMSCs in the treatment on POI and its potential mechanism, which may significantly impact the field of reproductive health. We constructed a rat model of early-onset ovarian dysfunction that closely mimics the physiological and pathological state of human POI. This allowed us to accurately evaluate the positive effects of PMSCs and their related factors in improving POI symptoms and promoting ovarian function recovery. The insights and methods we present in this study for the treatment of POI and the promotion of female reproductive health are a beacon of hope, laying a solid foundation for future women's health research and practice.

## Materials and methods

### Animals

Healthy female Sprague-Dawley (SD) rats (6-8 weeks old, weight $240 \pm 10\,g$) were purchased from the Experimental Animal Center of Ningxia Medical University. All animal experiments were conducted with every effort to minimize animal suffering and were approved by the Ethics Committee of Ningxia Medical University (IACUC No. Iacuc-nylac-2023-094). The animals were kept in a temperature-controlled chamber for a week (temperature maintained at $22 \pm 2°C$ and air humidity controlled at 40-70%) and subjected to a 12-hour cycle of light and darkness in a polycarbonate cage. The feed was purchased from Beijing Gaoli Feed Co., LTD. (crude protein 46.65%, moisture 20.73%, crude fat 0.09%, crude ash 0.13%, crude fiber 0.07%, trace elements calcium and phosphorus).

### Isolation and culture of human placental mesenchymal stem cells (PMSCs)

We collected full-term placenta tissue from healthy mothers in strict compliance with routine elective cesarean section procedures at Ningxia Medical University General Hospital. Our meticulous process ensures the high quality and purity of the research materials. We

guarantee that all research activities are based on the explicit informed consent of the mother and that the research protocol has been approved by the Human Research Ethics Committee of Ningxia Medical University (IACUC No. KYLL-2023-0269) to maintain the legitimacy and transparency of the research. The collected placental tissue was finely trimmed by ophthalmic scissors, avoiding blood vessels and accurately obtaining the chorionic tissue. Subsequently, the obtained placenta tissue was thoroughly rinsed more than three times with clean phosphate-buffered saline (PBS, 0.0067 M PO4, pH 7.0-7.2) to ensure clean and contaminant-free. The rinsed tissue is finely broken by mechanical action and then transferred to the centrifuge tube, DMEM culture-medium (GIBCO, Thermo-Fisher Scientific, USA), collagenase type A of 1 mg/mL(GIBCO, Thermo-Fisher Scientific, USA), and DNase enzyme of 0.05 mg/mL(GIBCO, Thermo-Fisher Scientific, USA). The tissues were digested for two hours at a constant 37°C water bath. Once this was done, we removed the waste liquid by centrifugation, resuspended the cells with PBS, and filtered them step by step using a two-stage cell sieve filtration system (100 μm vs. 40 μm cell sieve) to obtain a pure cell suspension. After centrifuging the collected cell suspension, we used a serum-free mesenchymal stem cell culture medium (Yinfeng Biological Group Co. Ltd, Jinan, Shandong Province, China) for re-suspension. The cells were inoculated in a 10 cm sterile petri dish and cultured under constant conditions of 37°C and 5% $CO_2$. To ensure cell quality and activity, we change the culture medium every two days. The first cultured cells were defined as primary cells [19]. When the cells had grown to a 70-80% fusion state, we performed cell passage using TrypLE™ Express (1x concentration, GIBCO, Thermo-Fisher Scientific, USA). This paper used the fifth-generation PMSCs as experimental materials in the subsequent cell therapy experiments.

## Identification and differentiation of PMSCs

First, cells cultured to the third generation were digested and collected with TrypLE™ Express. Next, we adjusted the cell concentration to $1 \times 10^6$ cells/mL using a PBS suspension. Subsequently, we added PE-labeled CD45, CD73, CD90, and CD105 antibodies and FITC-labeled HLA-DR, CD34, and CD14 antibodies (BD, USA) to test tubes containing 200 μL cell suspension, strictly following the manufacturer's operating guidelines. To ensure the accuracy of the results, we also set up a negative control group. All test tubes were incubated at room temperature for 30 min away from light, and the cells were washed with PBS and then detected by flow cytometry.

We inoculated human placental mesenchymal stem cells cultured to the 5th generation into a 6-well plate, ensuring that the cell density per well was $1 \times 10^5$ cells/mL. When the degree of cell fusion approached 80%, we initiated the differentiation process. According to the product description, osteogenic, lipogenic, and chondrogenic differentiation fluids were used to induce the differentiation of these cells. At the same time, control cells continued to be added to the mesenchymal stem cells-specific culture medium. After 21 days of induced differentiation, the cells in the differentiated group were meticulously fixed and stained. First, the cells were fixed at room temperature for 30 min using 4% paraformaldehyde. We then stained the cells with three specific staining agents, alizarin red, oil red O, and Yasin blue, respectively, to distinguish adipoblast, osteoblast, and chondroblast. Finally, they were carefully observed under a microscope and photographed to verify the effect of induced differentiation on different cell types.

## POI model construction and PMSC treatment experimental animal grouping

As shown in Fig 1, 40 female SD rats aged 6 to 8 weeks were selected for this study. After one week of adaptive feeding, they were equally divided into four groups with ten rats in each group.

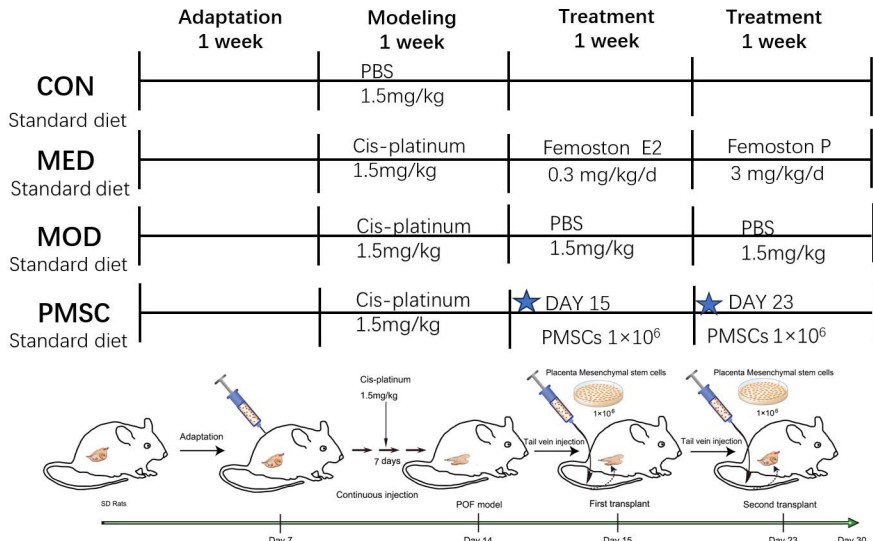

**Fig 1. Flow chart of animal experiment.**

The use of a random number table method for group division ensured the fairness and objectivity of the study. The groups were as follows: the CON group, serving as the control group, received the same amount of PBS intraperitoneal injection as the other groups, providing a baseline for comparison. The MOD group, as the POI (premature ovarian failure) animal model group, received an intraperitoneal injection of 1.5mg/kg cisplatin solution daily for seven consecutive days. After POI modeling, the MED group was given 0.3mg/kg/d of estradiol and 3mg/kg/d of dydrogesterone for 14 consecutive days according to Femoston instructions.

The PMSC group was used to study the treatment of cisplatin-induced POI models with PMSCs, a promising regenerative therapy due to their immunomodulatory and tissue repair properties. After establishing the POI model, the rats' body weight and estrus cycle were diligently monitored throughout the experiment. The POI model was evaluated by measuring ovary weight, steroid hormone level, and morphological observation.

### Estrous cycle monitoring and ovarian follicle count in rats

Vaginal smears of rats were collected daily at 9:00 am, and the stages of the estrous cycle were determined by Wright-Giemsa staining, which was performed by microscopically analyzing the dominant cell types in the smears. The estrus cycle consists of four stages [20], composed of different dominant cells: the preestrus was mainly composed of nucleated epithelial cells; In the estrus period, the keratinoid epithelial cells were dominant. The late stage of estrus contains white blood cells, nucleated epithelial cells, and keratinized epithelial cells. White blood cells were the main cell types of interest. After sampling, the ovarian tissue samples were immediately fixed with 4% paraformaldehyde (Servicebio, Wuhan Servicebio Technology Co., LTD., Wuhan, Hubei, China), then paraffin-embedded and cut into 4μm slices. To assess ovarian damage, ten sections (total n = 8) were selected from each sample placed on glass slides and treated with hematoxylin and eosin (H&E) staining (Solarbio, Beijing, China). Subsequently, two experts, unaware of the sample source, used an Olympus optical microscope (Melville, NY, USA) to analyze the follicles (primary, secondary, mature, and atretic) in the section classified and counted [21]. The percentage of follicles at each stage was calculated and compared between groups.

## Rat sample collection and handling

Our experimental protocol, meticulously designed to ensure the utmost accuracy of the results, is of paramount importance to our work. The rats in each group were required to fast for 12 hours after the last dose, and then to be anesthetized by inhaling isoflurane. During anesthesia, we continuously monitored the rats' respiratory status and corneal reflexes, demonstrating our commitment to their welfare and the reliability of the experimental results. Once the rat was successfully anesthetized, we placed it in a supine position and immobilized its limbs. The skin of the chest and abdomen was thoroughly disinfected and the heart was carefully exposed with surgical instruments for cardiac blood extraction. After blood collection, the blood in the collection vessel was left to rest for 30 min and centrifuged for 10 min at 4°C at $12000 \times g$. After centrifugation, we removed the supernatant and subpack to label it. To maintain the activity of the sample, we quickly froze it in liquid nitrogen, and then transferred it to the -80°C refrigerator for storage for subsequent experiments. After the blood collection procedure was completed, bilateral ovarian tissue was further collected. One ovary was immobilized in 4% paraformaldehyde for subsequent HE staining and immunohistochemical analysis. The other ovary was rinsed with normal saline (Shandong Qidu Pharmaceutical CO., LTD.) to remove excess tissue, such as fat and fallopian tubes, leaving only the ovarian tissue itself. The tissue was then placed in a centrifuge tube and immediately transferred to a -80°C ultra-low temperature refrigerator for further use.

## Determination of steroid sex hormones

To accurately assess steroid sex hormone levels in rats, we followed the instructions of Proteintech (Wuhan, China) enzyme-linked immunosorbent assay (ELISA) kit and performed detailed tests on collected plasma samples. Specifically, we measured concentrations of follicle-stimulating hormone (FSH), luteinizing hormone (LH), estrogen (E2), testosterone (T), progesterone (PROG), and prolactin (PRL); the detection sensitivities of these indexes were 0.1 IU/L, 0.1 mIU/ml, 0.1 pmol/L, 1.0 pg/mL, 0.1 nmol/L and 0.1 ng/mL, respectively. To ensure the accuracy and reliability of the data, we conducted three independent measurements for each sample with utmost precision. By measuring each pore's absorbance value (OD value) at 450 nm and combining it with the standard curve, we calculated the exact levels of FSH, LH, E2, T, PROG, and PRL in rat plasma.

## Plasma and ovarian inflammatory markers and determination of LPS

The measurements of inflammatory cytokines in plasma and ovary, including IL-1β, IL-6, IL-10, IL-33, and TNF-α, were conducted with thoroughness and attention to detail using ELISA kits provided by Proteintech (Wuhan, China). The sensitivity of IL-1β, IL-10, and IL-33 was 0.1 pg/mL, and the sensitivity of IL-6 and TNF-α was 1.0 pg/mL, respectively. Each sample was independently measured three times to ensure the accuracy of the results. In addition, the plasma LPS levels in each group were measured using Limulus reagents provided by Xiamen Baiyuan Technology Co., LTD. (Xiamen, China). The procedure was meticulously followed: 50 μL of diluted plasma (diluted with endotoxin-free water in a ratio of 1:4) was added to each hole of the 96-well plate, and then 50 μL of Limulus reagent was added at the initial time point. After incubation at 37°C for 30 min, 100 μL of chromogenic substrate was added to each well and continued incubation at 37°C for 6 min. Finally, a termination solution was added to stop the reaction, and the optical density values were measured at 545 nm using a microplate reader, ensuring the validity and reliability of the results.

## Intestinal microbiota analysis

Sample collection and pretreatment: At 24 hours before the start of the experiment, six rats were randomly selected from each group and transferred to separate sterile cages to collect their fresh feces. Immediately after collection, the fresh feces were frozen in liquid nitrogen and stored at −80°C for subsequent DNA extraction. To extract total DNA, first weigh 0.5 g of feces into 2 mL centrifuge tubes, add 1 mL of SLX-Mlus Buffer and 500 mg of magnetic beads, and grind using a grinder at 250 seconds and 45 Hz parameter setting. The total DNA was then extracted according to the instructions on the E.Z.N.A.® Soil DNA Kit. To verify the quality of the DNA, we used 1% agarose gel electrophoresis and measured the concentration and purity of the DNA with a Nanodrop 2000 c U.V.-vis Spectrophotometer. PCR amplification targets the V3 and V4 hypervariable regions of the 16S rRNA gene analysis using specific primers (338F-ACTCCTACGGGAGGCAGCAG and 806R-GGACTACHVGGGTWTCTAAT) [22]. The PCR reaction mixture including 4 μL 5 × Fast Pfu buffer, 2 μL 2.5 mM dNTPs, 0.8 μL each primer (5 μM), 0.4 μL Fast Pfu polymerase, 10 ng of template DNA, and ddH$_2$O to a final volume of 20 μL. PCR amplification cycling conditions were as follows: initial denaturation at 95°C for 3 min, followed by 27 cycles of denaturing at 95°C for 30 s, annealing at 55°C for 30 s and extension at 72°C for 45 s, and single extension at 72°C for 10 min, and end at 4°C. PCR product processing and sequencing: To confirm the quality and purity of PCR products, we used 2% agarose gel electrophoresis. Subsequently, an equal volume of 1 × loading buffer (including SYB green) was mixed with PCR products, and agarose gels containing target DNA were purified using the Qiagen Gel Extraction Kit. After purification, the PCR products were quantified using Quantus™ Fluorometer (Promega, Madison, Wisconsin, USA). Libraries were generated by the TruSeq DNA PCR-Free Sample Preparation Kit (Illumina, San Diego, USA), and whole genome sequencing was carried out using the Illumina platform NovaSeq sequencer to obtain 250 bp peer read data. Data analysis: The received data is first subjected to strict quality control by Fastp software and sequentially spliced by FLASH software. Then, using UPARSE software, we performed OTU cluster analysis of the non-redundant sequences with 97% similarity. In this process, we use UCHIME software to eliminate potential chimera sequences and use the RDP classification method combined with the Silva database (SSU132) for detailed species classification annotation of sequences. This thorough data analysis process ensures the validity of our findings. These data will be used for OTU relative abundance analysis, α and β diversity assessment, and comparison of typical and endemic OTUs between samples to study differences in community composition across samples at different taxonomic levels. Finally, we will use correlation analysis software to explore the potential relationship between strains and abnormal indicators. This research has the potential to inspire and motivate further studies in the field of microbiology and genetics.

## Determination of SCFAs in rat stool

We used the gas-mass spectrometry system (Thermo Scientific™ TRACE™ 1310) to accurately measure SCFAs in rat stool samples. In the process of chromatographic analysis, we selected the Agilent HP-INNOWAX capillary column (30 m × 0.25 mm, film thickness 0.25 μm), and the sample was injected with 1 μL sample volume and 10:1 shitter ratio. The inlet and transmission line temperature is 250°C, and the ion source temperature is 300°C. In terms of temperature control, we use a programmed temperature rise method: the initial temperature of 90°C rises to 120°C at 10°C/min, then rises to 150°C at 5°C/min, and finally rises to 250°C at 25°C/min quickly and maintains it for 2 min. In the whole process, the carrier gas is helium with a flow rate of 1.0 mL/min. For mass spectrometry, we selected

a selective ion monitoring (SIM) scanning mode, combined with an electron bombardment ionization (EI) source with an electron energy of 70 eV, to ensure the acquisition of high-quality mass spectrometry data. To ensure measurement accuracy, we configured standard curves involving precisely weighed pure standards such as acetic acid, propionic acid, butyric acid, isobutyric acid, valeric acid, isovaleric acid, and caproic acid. We formulated a series of mixed standard solutions with a concentration gradient using ether. These standard solutions are stored at 0°C and used to draw standard curves. The curve is based on the linear regression relationship between the ratio of the concentration of the standard substance and the corresponding peak area, and the linear regression equations for all substances show a high correlation (correlation coefficient > 0.99). In addition, we verified the stability and reliability of the measurement method by evaluating precision, recovery rate, repeatability, limit of quantitation, and QC quality control results. Metabolite extraction and calculation: In the metabolite extraction phase, we first moved a precisely weighed fecal sample into a 2 mL centrifuge tube and added 50 μL of 15% phosphoric acid, 100 μL of internal standard (isocaproic acid) solution with a concentration of 125 μg/mL, and 400 μL of ether. After homogenization for 1 min and centrifugation for 10 min (12000 × g, 4°C), the supernatant was taken for machine test. Finally, the SCFAs content was calculated according to the following formula: Content (μg/g) = (C × 0.5)/sample size × 1000, where C unit is μg/mL and sample size unit is mg.

## Statistical analysis

GraphPad Prism 7 (GraphPad Software, San Diego, California, USA) was utilized. The data were considered significant at $p < 0.05$ after using the one-way analysis of variance (ANOVA), followed by Tukey's post hoc statistical test. The results were presented as mean±standard deviation (SD).

# Results

## Culture and identification of PMSCs *in vitro*

After isolation and purification, PMSCs extracted from fresh placenta showed unique cell morphology and excellent differentiation potential. These cells showed the characteristics of adhesive growth, the shape of a long spindle, tightly intertwined in a swirl arrangement, and the cytoplasm filled. The nucleoli was visible, arranged, parallel, or rotated, morphologically similar to fibroblasts (Fig 2A). Under specific induction conditions, PMSCs exhibited the ability to differentiate into a variety of cell types. Among them, after 21 days of lipid-induced culture, oil red O staining showed the formation of significant bright red lipid droplets in the cells, indicating that PMSCs had successfully differentiated into adipocytes (Fig 2B). After 21 days of culture under osteogenic induction conditions, alizarin red staining revealed the deposition of a large amount of red mineralized matrix, indicating the potential of PMSCs to differentiate into osteoblasts (Fig 2C).

Similarly, when PMSCs were cultured under chondroblast-induced conditions for 21 days, Asin blue staining showed that all cells were blue because the cells expressed type II collagen, which is specific to chondrocytes, confirming their ability to differentiate into chondroblasts (Fig 2D). Combined with these experimental results, we could confirm that PMSCs isolated from placental tissues were capable of differentiation into adipocytes, osteoblasts, and chondroblasts in three ways. Flow cytometry analysis further revealed that mesenchymal stem cell markers CD105, CD90, and CD73 were highly expressed on the surface of these PMSCs, while hematopoietic stem cell markers CD34, CD45, monocyte marker CD14, and HLA-DR were at very low or no expression on the cell surface (Fig 2E).

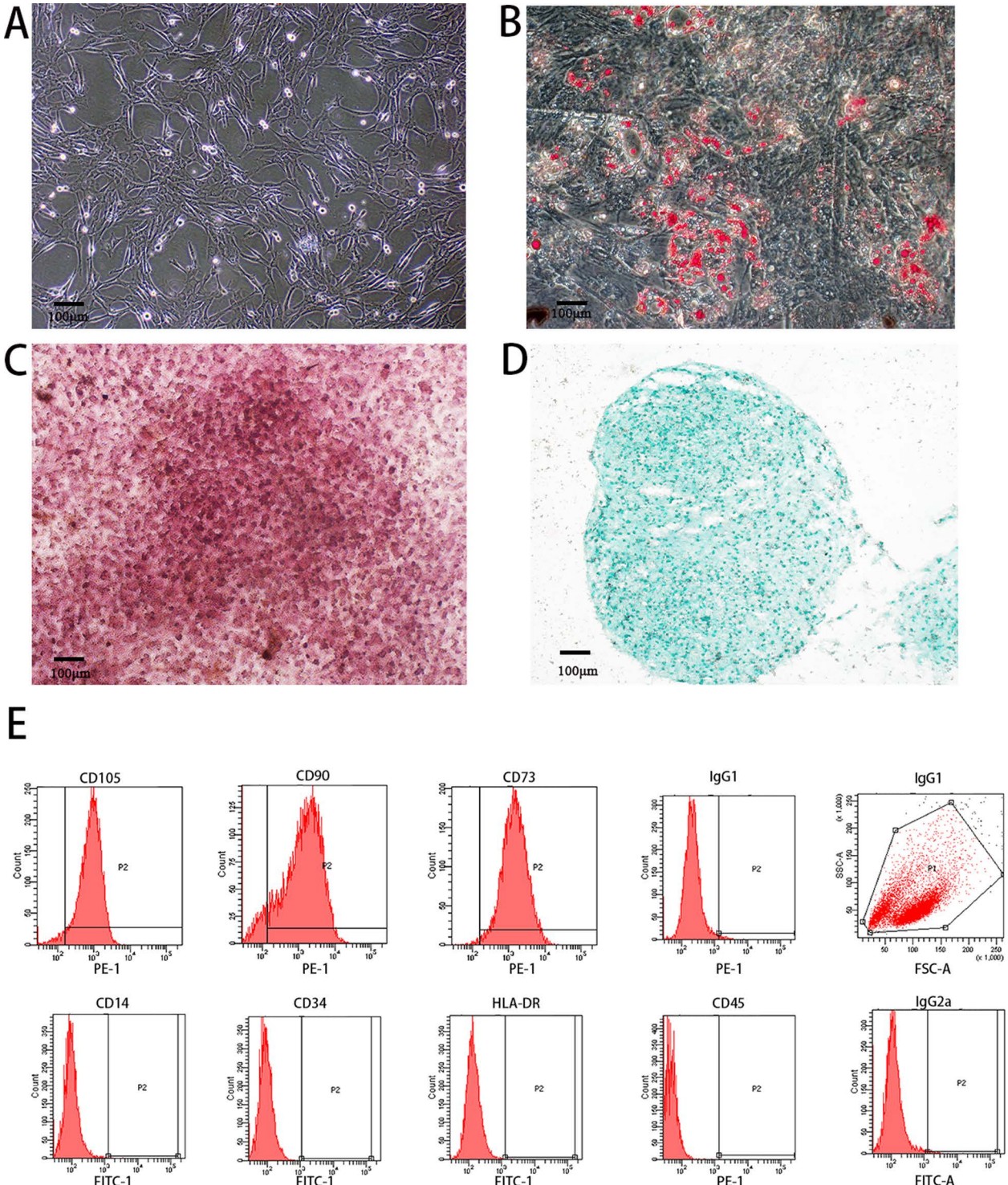

**Fig 2. Morphological observation and identification of PMSCs.** A: PMSCs culture; B: PMSCs lipogenesis induced oil red O staining; C: PMSCs osteogenesis induced alizarin red staining; D: PMSCs chondrogenesis induced Yasine blue staining; E: PMSCs Surface Marker detection (A-D:200x).

## General state, body weight, and food intake of rats in each group

Beginning with the adaptive feeding phase, we measured the rats' weight regularly every other day. The results showed that the rats in the CON group had good general health, entire mental state, bright hair, and average daily eating and excretion. The MOD rats, on the other hand, showed an unhealthy state, with slow movement and dark, yellowish hair. Before modeling, all groups had no significant difference in body weight (P > 0.05). However, statistical analysis of body weight after modeling showed that compared to the CON group, rats in the MOD group showed a significant decrease in body weight (P < 0.0001, Fig 3A). To assess the feeding of the rats more accurately, we adopted a calculation method of feeding rations (500 grams per cage) and measuring the remaining amount after 24 hours. The results showed that the food intake of the MOD group was notably lower than that in the CON group (P < 0.0001, Fig 3B). The body weight and food intake after PMSCs intervention showed no significant difference.

## Ovarian weight and ovarian index in diverse groups

Compared with the CON and PMSC groups, the ovarian weights of rats in the MOD group were significantly reduced. At the same time, the ovarian weight of rats in the MED group showed decreased compared with the CON group (P < 0.01 in the MOD group versus the CON group, P < 0.05 in the MOD group versus the PMSC group, Fig 4A). By accurately calculating the ovarian index of rats, we observed that the ovarian index in the MOD group was significantly lower than that in the CON or PMSC groups (P < 0.01 in the MOD group vs. the CON and PMSC groups, Fig 4B, 4C). These accurate and precise findings suggested that intervention with PMSCs significantly improved ovarian weight and index in POI (primary ovarian insufficiency) rats, showing its potential therapeutic effect.

## PMSCs intervention rectified the estrous cycle of rats with POI

We compared the differences in the estrous cycle of each group of rats by a vaginal smear assay using Wright-Giemsa staining. The results showed that the rats in the CON group presented a standard estrus cycle, including pre-estrus, estrus, post-estrus, and inter-estrus, and the whole cycle lasted steadily for 4-5 days, as shown in Fig 5A–5E. However, after cisplatin-induced POI, the estrous cycle of MOD group rats showed significant interference, which was characterized by irregular cycles and mainly stayed in the late estrous period (Fig 5G). It is worth noting that after the intervention of PMSCs or estrogen drugs, we observed that the estrous cycle of these two groups of rats gradually restored to a complete and regular cycle state in the late intervention period (Fig 5F and 5H).

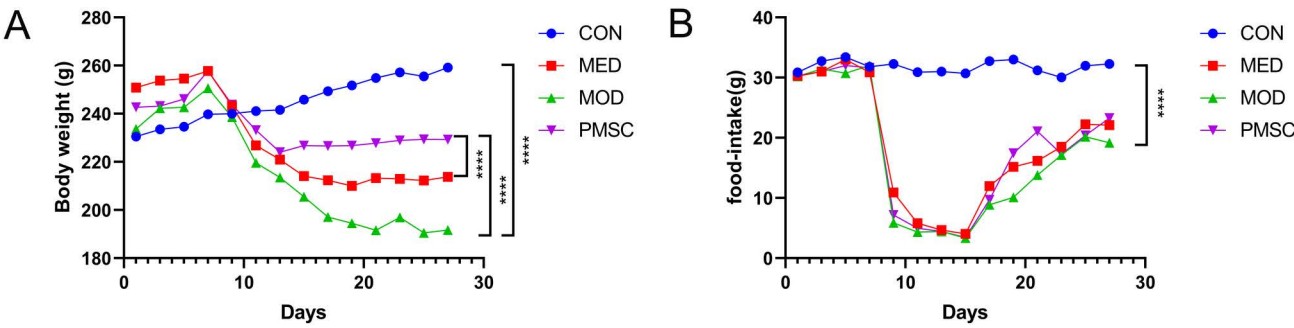

**Fig 3. Comparison of body weight and food intake of rats in each group.** A: Analysis of body weight difference among each group; B: Comparison of food intake difference among all groups (****p < 0.0001).

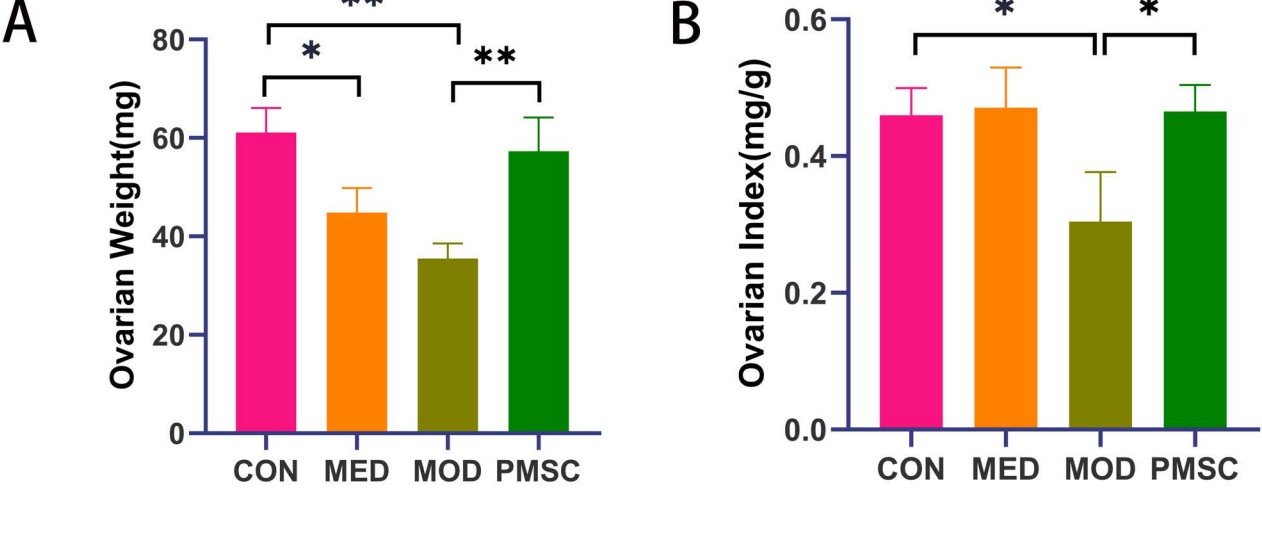

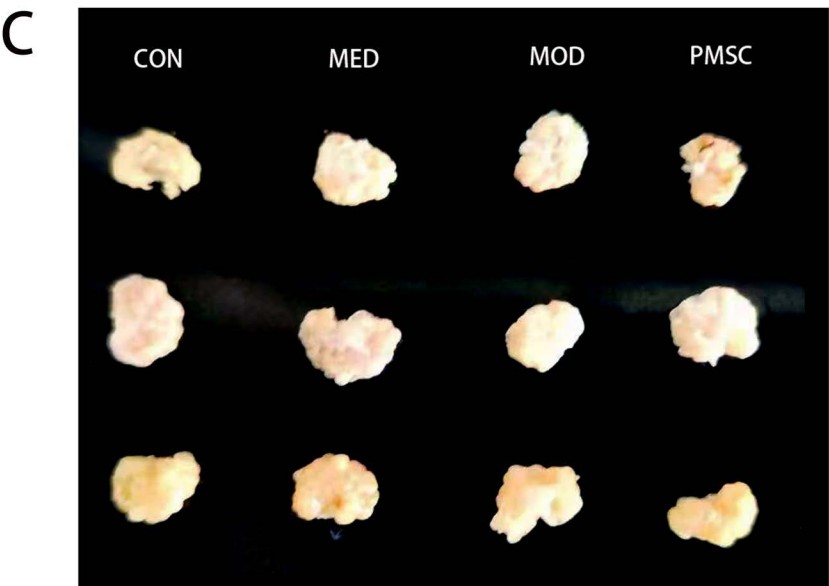

**Fig 4. Comparison of ovarian weight and ovarian index of rats in each group.** A: Comparison of ovary weight in each group; B: Comparison of ovarian index in each group; C: unilateral ovarian tissue (*$p < 0.05$, **$p < 0.01$).

## PMSCs intervention attenuated the pathological damage of ovarian tissue in rats with POI

The HE staining was used to observe and analyze the pathological changes of ovaries in each group in detail (Fig 6A–6D). In the CON group, the follicles within the ovarian tissue showed normal morphology at different stages of development. However, in the MOD group, the ovarian tissue structure showed significant disorder, as the number of primary, secondary, and mature follicles decreased significantly ($P < 0.001$, Fig 6C, 6E). In contrast, the atretic follicles increased ($P < 0.05$, Fig 6C, 6E). These changes indicated the impaired ovulation in the cisplatin-induced POI model. However, the study brings a ray of hope as it is gratifying to note that the ovarian tissue structure of the rats was significantly improved after treatment

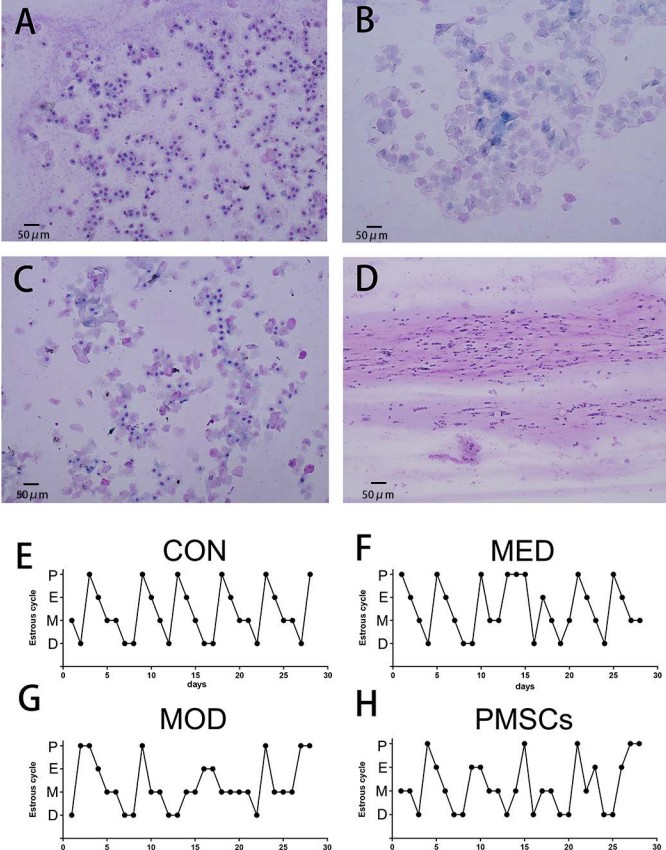

**Fig 5. Comparison of the estrus cycle of rats in each group.** A: vaginal smear staining during pre-estrus; B: Vaginal smear staining during estrus; C: vaginal smear staining during post-estrous; D: vaginal smear staining during inter-estrus; E: estrous cycle of CON group; F: estrous cycle in MED group; G: estrus cycle of MOD group; H: estrous cycle of PMSC group (A-D:200×).

with PMSCs. Specifically, primary, secondary, and mature follicles increased significantly ($P < 0.05$). In contrast, the number of atretic follicles decreased significantly ($P < 0.01$), indicating that PMSCs could dramatically improve the abnormal status of ovarian follicles in POI rats. This potential of PMSCs offers a promising outlook for the treatment of POI.

## PMSCs intervention ameliorated the homeostasis of steroid hormones in rats with POI

Six steroid sex hormones including follicle-stimulating hormone (FSH), luteinizing hormone (LH), estrogen (E2), testosterone (T), progesterone (PROG), and prolactin (PRL), were accurately quantified. After statistical analysis, we found that FSH levels in the MOD group were significantly increased compared with those in the CON group ($P < 0.01$). In contrast, FSH levels were significantly decreased after PMSCs intervention ($P < 0.05$, Fig 7A). At the same time, the E2 level in the MOD group was considerably lower than that in the CON or MED groups ($P < 0.05$, Fig 7B). Although there was no statistically significant difference in E2 level between the MOD group and the PMSC group ($P = 0.0534$, Fig 7B), there was a trend difference between the two groups. In addition, E2 level in the MOD group was also lower than that in the MED group ($P < 0.05$, Fig 7B). Another notable finding was that T level was elevated

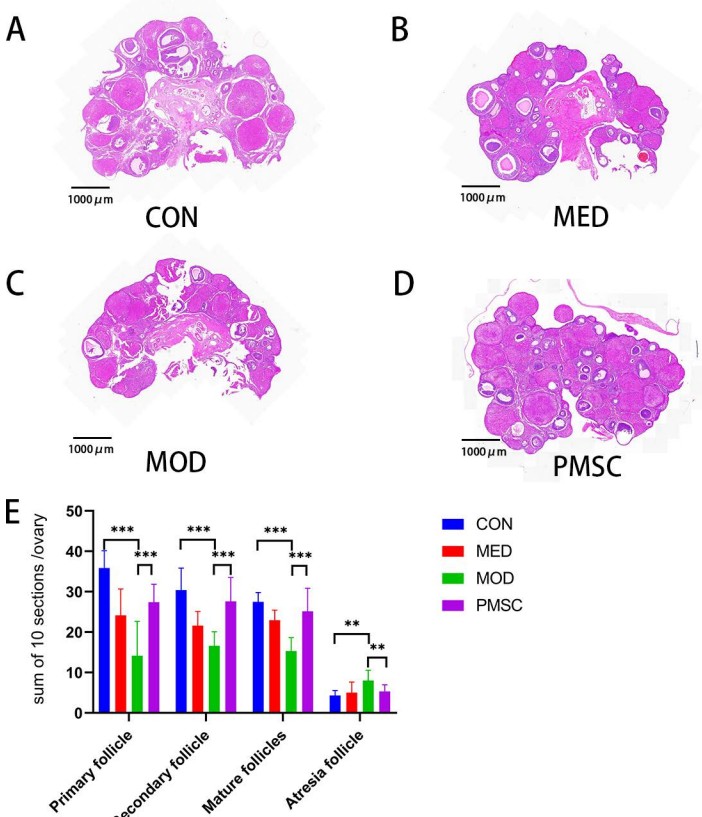

**Fig 6. Comparison of HE staining results and the number of follicles at different levels in ovarian tissue of rats in each group.** A: The hematoxylin-eosin (H&E) stain of CON group ovarian tissue; B: The hematoxylin-eosin (H&E) stain of ovarian tissue in MED group; C: The hematoxylin-eosin (H&E) stain of MOD group ovarian tissue; D: The hematoxylin-eosin (H&E) stain of ovarian tissue in PMSC group; E: Comparison of the number of follicles at all levels in each group (A-D:40×, *$p < 0.05$, **$p < 0.01$, ***$p < 0.001$).

in the MED group compared to the MOD group ($P < 0.05$, Fig 7C). However, no significant difference was found in the levels of PROG and PRL among the groups ($P > 0.05$, Fig 7D, 7F). In addition, LH level in the MOD group was significantly lower than that in the CON and MED groups ($P < 0.01$ and $P < 0.05$, Fig 7E). These results suggested that PMSCs intervention significantly improved hormone levels in POI rats by predominantly regulating FSH and E2.

## Changes in levels of inflammation-related factors in each group

As shown in Fig 8, plasma level of proinflammatory cytokine IL-1β in the MOD group was significantly higher than that in the CON and MED groups (MOD vs CON $P < 0.01$, MOD vs MED $P < 0.001$, Fig 8A). Similarly, the level of IL-1β in ovarian tissue of the MOD group also exerted an increase compared with that in the CON group ($P < 0.01$) and MED group ($P < 0.05$, Fig 8A). In addition, IL-6 level in ovarian tissue of the MOD group was significantly elevated compared to the CON and MED groups ($P < 0.001$, Fig 8B). The level of the anti-inflammatory factor IL-10 in ovarian tissue was reduced in the MOD group and MED group ($P < 0.001$, Fig 8C). In plasma, TNF-α level was down-regulated only in the MED group ($P < 0.05$), but in ovarian tissue, TNF-α level was higher in the MED group than that in the CON group ($P < 0.001$, Fig 8D). It was noteworthy that the ovarian TNF-α level in the MOD group was significantly higher than that in the CON group ($P < 0.001$, Fig 8D). The plasma

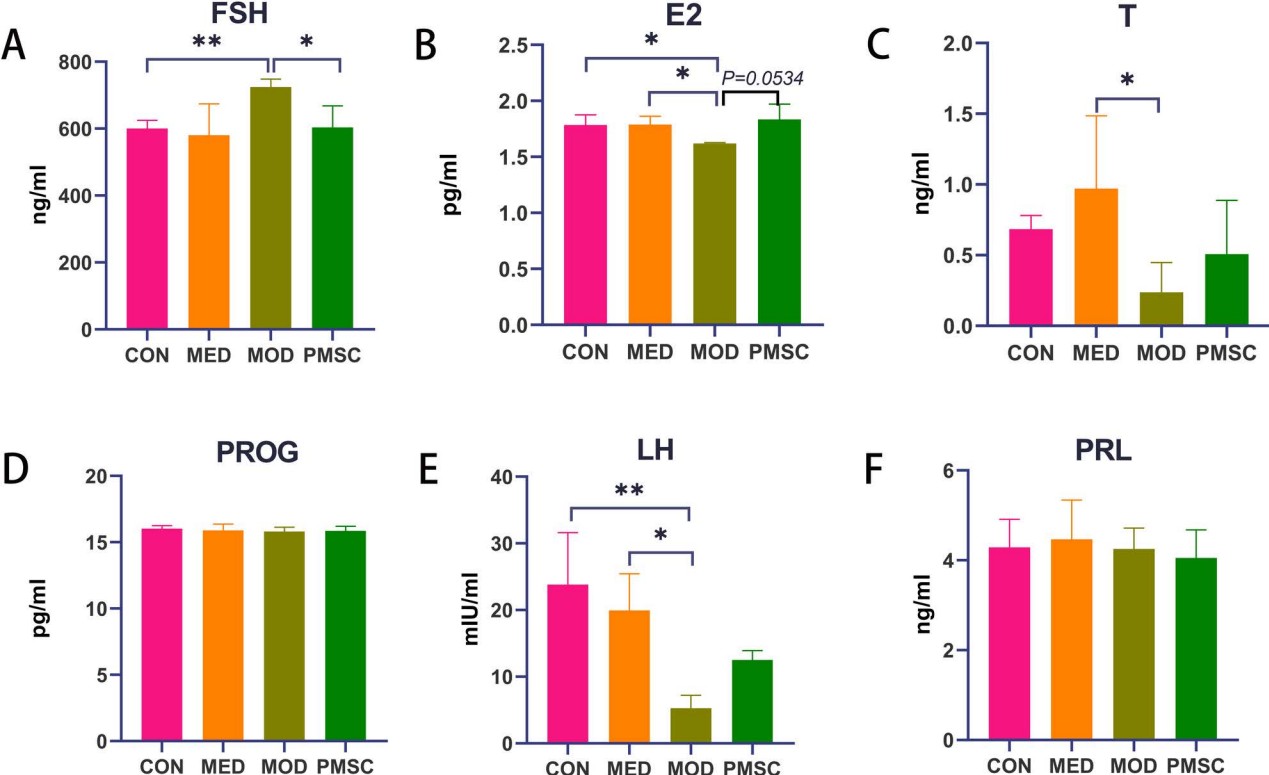

**Fig 7. Comparison of steroid hormone levels in rats in each group.** A: Follicle-stimulating hormone FSH level; B: Estrogen E2 level; C: Testosterone T level; D: Progesterone PROG level; E: luteinizing hormone LH level; F: Prolactin PRL level (*$p < 0.05$, **$p < 0.01$).

IL-33 level in the MOD group was also higher than that in the MED group (P < 0.05, Fig 8E). Notably, after intervention with PMSCs, multiple proinflammatory cytokines in POI rats were significantly inhibited, including IL-1β in plasma (P < 0.001, Fig 8A), IL-6 in ovarian tissue (P < 0.05, Fig 8B), and TNF-α in ovarian tissue and plasma (P < 0.01, Fig 8D), and plasma IL-33 (P < 0.05, Fig 8E). Meanwhile, the anti-inflammatory factor IL-10 in ovarian tissue was significantly up-regulated after the intervention (P < 0.01, Fig 8C). These results suggested that PMSCs significantly improved POI in rats by inhibiting the inflammatory response.

## PMSCs intervention improved the endotoxemia in rats with POI

To further investigate the effects of PMSCs on gut dysbiosis, particularly on gram-negative bacteria, permeability, and intestinal barrier integrity, the pathogenic bacteria-derived LPS that translocated from intestine to plasma was measured after PMSCs treatment. The results showed that the plasma LPS concentration in the MOD group was significantly higher than that in the CON group, indicating the presence of endotoxemia in POI rats (P < 0.05, Fig 8F). After PMSC intervention, LPS levels in peripheral blood showed a downward trend, suggesting that PMSCs may play a role by affecting intestinal flora.

## PMSCs intervention modulated the difference of overall community structure of intestinal microbiota

Subsequently, to study the effects of PMSCs on the structure and abundance of intestinal flora in POI rats, we applied 16S rRNA sequencing to conduct a comprehensive analysis of

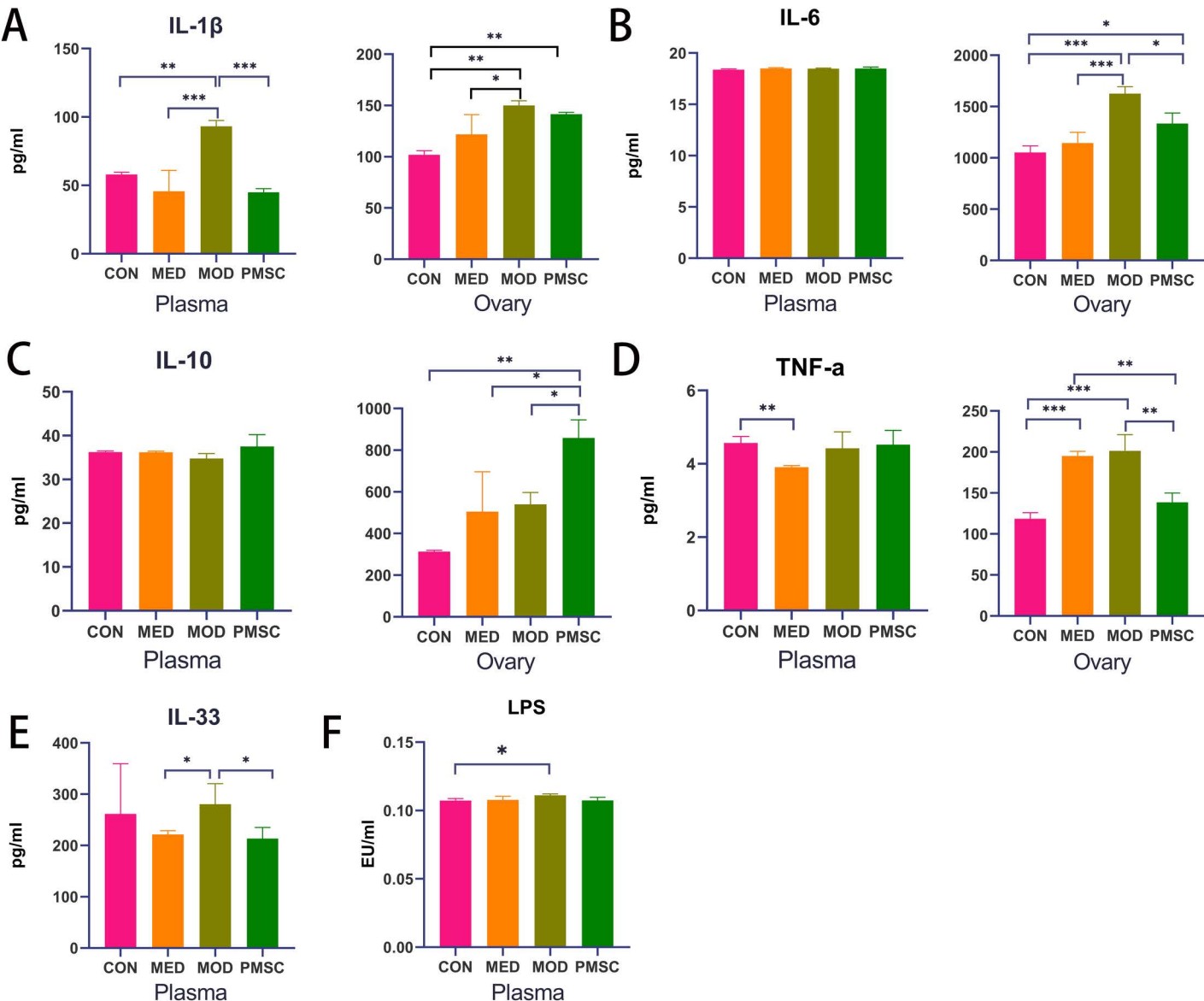

**Fig 8. Comparison of levels of inflammatory cytokines and LPS in each group.** A: IL-1β levels in plasma and ovarian tissue of rats in each group; B: IL-6 levels in plasma and ovarian tissue of rats in each group; C: IL-10 levels in plasma and ovarian tissue of rats in each group; D: Levels of TNF-α in plasma and ovarian tissue of rats in each group; E: Plasma IL-33 level of rats in each group; F: LPS levels in each group (*$p < 0.05$, **$p < 0.01$, ***$p < 0.001$).

fecal samples. First, we used sparse curves to evaluate the rationality of each group of rats' intestinal flora sequencing data. This evaluation method collects the α-diversity index of the corresponding samples by randomly sampling the sequences. It drew the α-diversity index curve that changes with the increased amount of extracted data. In this process, we pay special attention to Sobs (number of observed species) and Shannon index as evaluation indicators. The results showed that the data tended to be stable when the number of sequenced samples in each group reached 4000. This ensured that our sequencing data was sufficient to cover all microbial species in the samples with high reliability and rationality, as shown in Fig 9A.

Further analysis showed no significant difference in α diversity of gut microbiome by the Shannon index between the MOD and PMSC groups. However, community diversity was

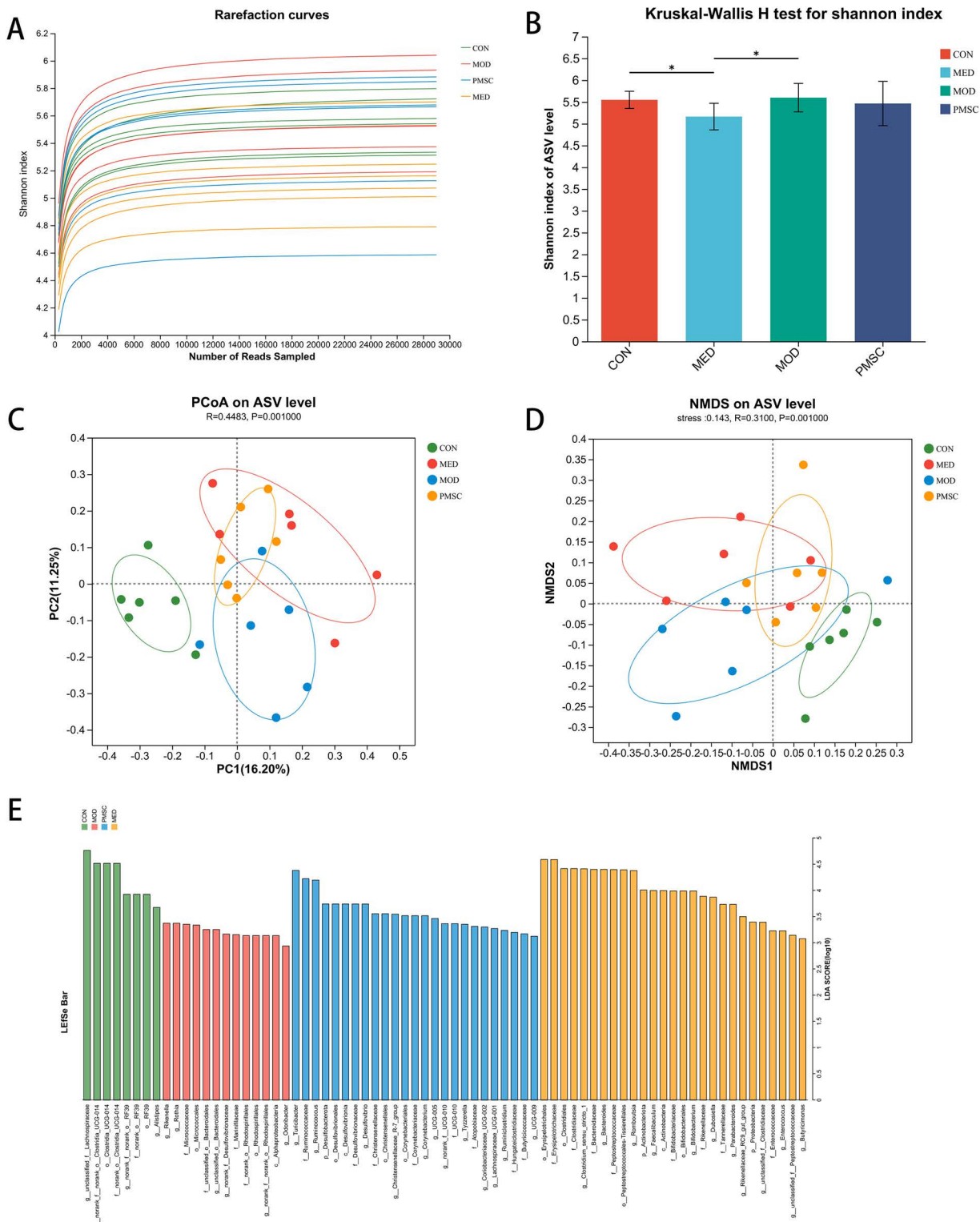

**Fig 9. Analysis of the difference in overall bacterial community structure in each group.** A: Dilution curves of 16S rRNA high-throughput sequencing groups were used to evaluate the rationality of sequencing data; B: α diversity of Simpson and Shannon indices; C, D: β diversity of PCoA based on weighted UniFrac distance and NMDS based on Bray Curtis; E: LEfSe analysis, LDA > 3. **N** = 5/ group; The data were expressed as mean ± SEM and analyzed by unpaired Student's t-test (*$p < 0.05$, **$p < 0.01$, ***$p < 0.001$).

significantly lower in the MED group than that in the CON and MOD groups (P < 0.05), and the difference was statistically significant (Fig 9B). To better understand the differences in gut microbial community structure among groups, we assessed β diversity using unweighted principal coordinate analysis (PcoA) and weighted distance matrix (non-metric multidimensional scale, NMDS). Both PcoA and NMDS analysis revealed significant differences in the intestinal microbial community structure of rats in different groups, and the clustering patterns among the groups were significantly separated (Fig 9C, 9D). In addition, to identify specific bacterial taxa that were closely related to each group, we performed LEfSe analysis. Analysis results showed that the MOD group was enriched in Rikenella, Micrococcaceae, Bacteroidales, and Desulfovibrionaceae, while the CON group enriched in Lachnospiraceae and Clostridia (Fig 9E). These findings provided important clues for us to understand further the effects of PMSCs on POI by modulating the intestinal microbiota. Specifically, PMSCs might exert their effects on POI by modulating the balance of these microbial communities, potentially reducing inflammation, improving intestinal barrier function, and enhancing motility. The enrichment of beneficial bacteria and suppression of harmful microbes could be key mechanisms through which PMSCs alleviate POI symptoms.

## The difference in intestinal flora in rats of each group at the phylum level

To further compare the differences among the diverse groups of gut microbiota, we conducted a detailed analysis of the gut microbiota at the phylum level. *Firmicutes* and *Bacteroidetes* dominate the intestinal bacterial community at the phylum level, accounting for about 90% of the entire bacterial community structure. In addition, *Actinobacteria* and *Desulfobacterota* also accounted for a certain amount, about 8%, while the abundance of other bacteria remained relatively low (Fig 10A). Specific to the experimental groups, the intestinal microbial community of the MOD group was *Bacteroidetes* as the main dominant bacteria, while the CON group was significantly inclined to *Firmicutes* (Fig 10B). Meanwhile, compared to the CON group, in the MOD group, the proportions of *Actinomyces* (P < 0.01), *unclassified_k_norank_d_Bacteria* (P < 0.05), *Proteobacteria* (P < 0.05), *Cyanobacteria* (P < 0.05), *Desulphurizing bacteria* (P < 0.05) and *Deferrobacteria* (P < 0.05) were significantly increased. This showed that cisplatin significantly affected the composition and structure of the intestinal flora in rats. Further, compared with the MOD group, the flora distribution of *Firmicutes*, *Bacteroides*, and *Proteobacteria* in the PMSC group showed significant differences (P < 0.05, Fig 10C, 10D), indicating that the intestinal flora composition of the PMSC group was more similar to that in the CON group. However, there was no significant difference in the distribution of *Firmicutes* and *Campilobacterota* between the MED group and the PMSC group (P > 0.05, Fig 10E). These findings provided valuable insights into understanding the effects of cisplatin on gut microbiota and the potential regulatory role of PMSCs.

## The difference of intestinal flora in rats of each group at genus level

On the genus level, there were differences in microflora among all groups (Fig 11A).Compared to the CON group, the proportions of *Unclassified_f_Lachnospiraceae* (P < 0.05), *norank_f_norank_o_Clostridia_UCG-041* (P < 0.05) and *norank_f_norank_o_RF39* in the MOD group were decreased (P < 0.05). The relative abundance of 8 species with relatively small proportions, such as *Bacteroides* (P < 0.05), *Bifidobacterium* (P < 0.01), *unclassified_k__norank_d__Bacteria* (P < 0.05), *norank_f_UCG_010* (P < 0.05) was significantly increased (Fig 11B).The results showed that POI rats had dysregulation of intestinal flora. After PMSC intervention, it was significant microbial community differencesat the genus level. Specifically, the abundance of *Turicibacter* and *Desulfovibrio* in the PMSC group is significantly increased.

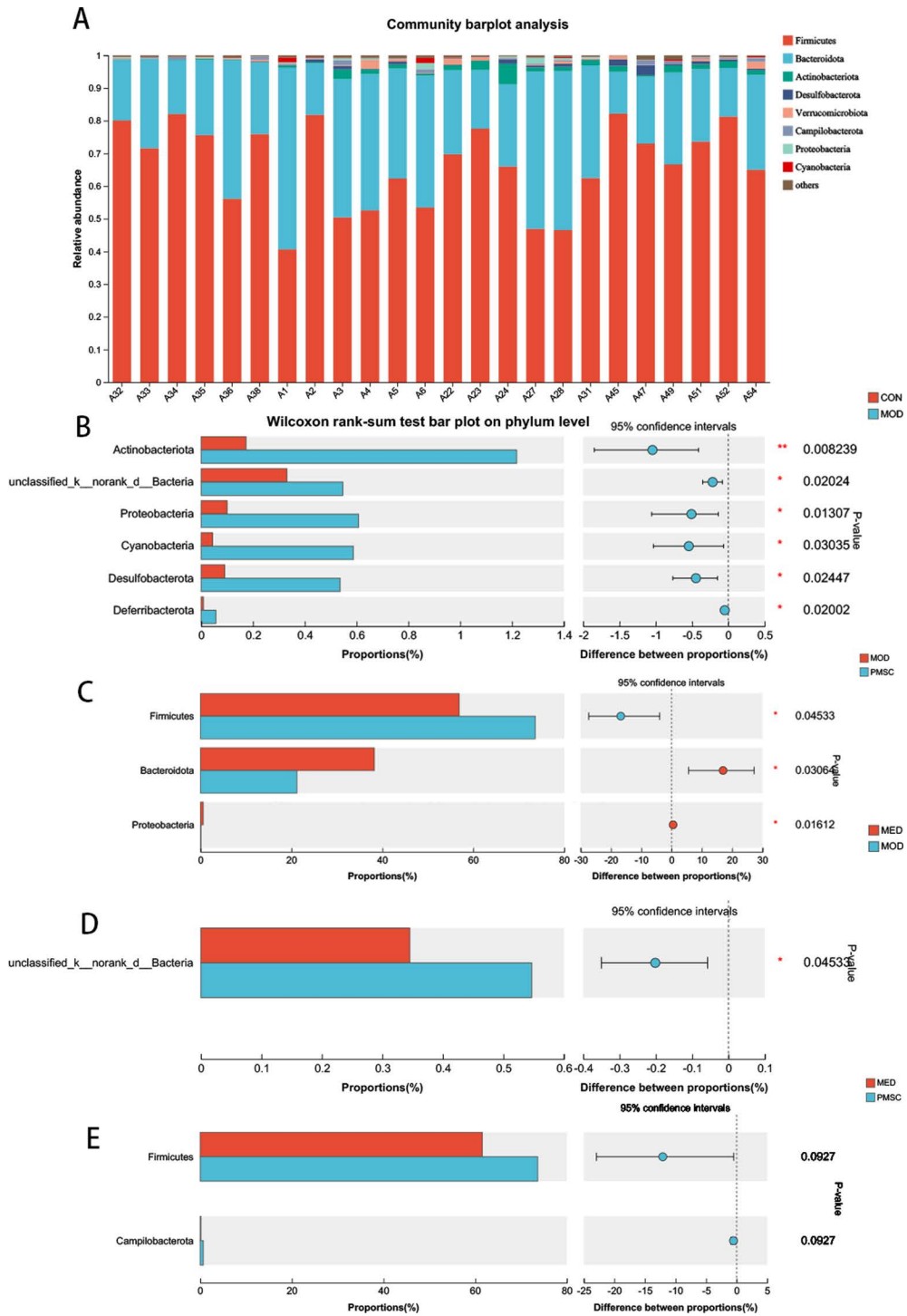

**Fig 10. The relative abundance of different microflora at intestinal portal level of rats in each group was different.** A: Relative abundance of microbial species at phyla level; B: Comparison of intestinal microbial species abundance in CON/MOD group; C: Comparison of intestinal microbial species abundance in MOD/PMSC groups; D: Comparison of intestinal microbial species abundance in MED/MOD group E: Comparison of intestinal microbial species abundance in MED/PMSC group (*$p < 0.05$, **$p < 0.01$, ***$p < 0.001$).

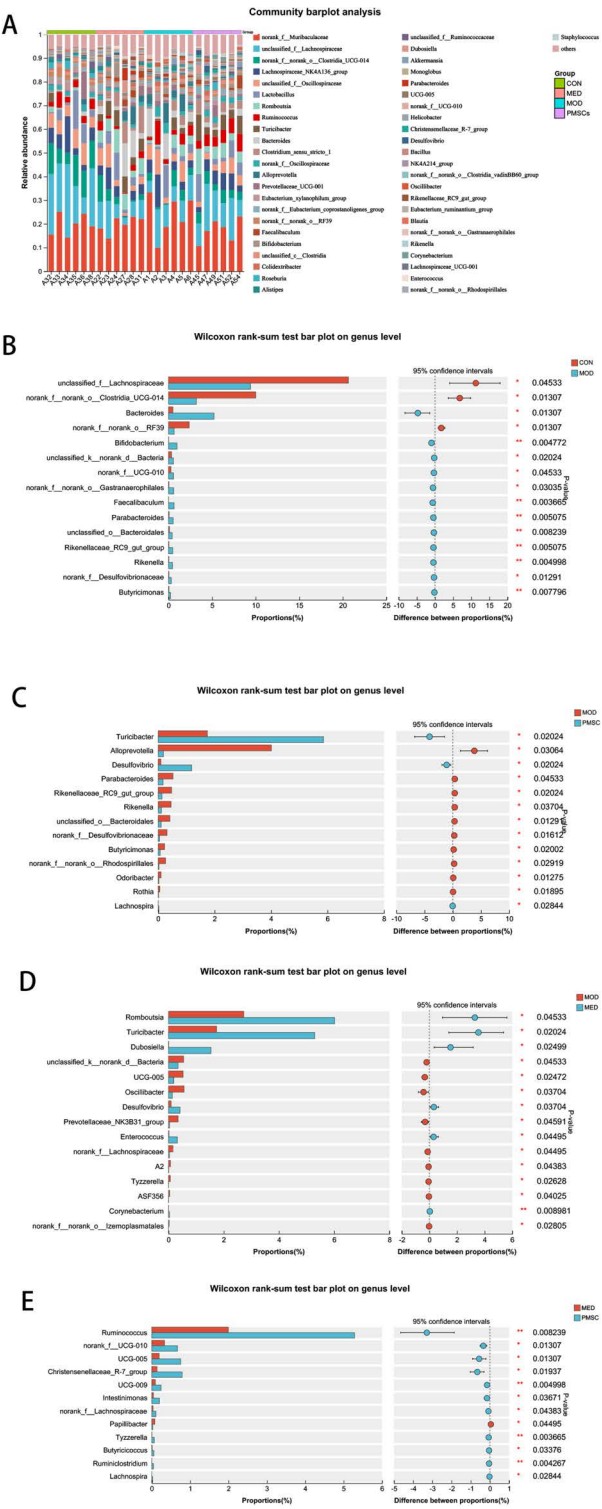

**Fig 11. Relative abundance of different intestinal flora of rats in each group at the genus level.** A: Relative abundance of microbial species at genus level B: comparison of intestinal microbial species abundance in CON/MOD group C: Comparison of intestinal microbial species abundance in MOD/PMSC group D: Comparison of intestinal microbial species abundance in MED/MOD group E: Comparison of intestinal microbial species abundance in MED/ PMSC groups (*$p < 0.05$, **$p < 0.01$, ***$p < 0.001$).

However, *Alloprevotella*, *Parabacteroides*, *Rikenellaceae_RC9_gut_group* and *Rikenella* were decreased (P < 0.05, Fig 11C). In addition, the abundances of *Romboutsia* (P < 0.05), *Turicibacter* (P < 0.05) and *Dubosiella* (P < 0.05) in MOD group were lower than those in MED group. The relative abundance of *unclassified_k__norank_d__Bacteria* (P < 0.05), *UCG_005* (P < 0.05), and *Oscillibacter* (P < 0.05) were increased compared with those in MED group (Fig 11D). The relative abundance of *Ruminococcus* (P < 0.01), *norank_f_UCG_010* (P < 0.05), *UCG_005* (P < 0.05), *Christensenellaceae_R-7_group* (P < 0.05) were lower in the MED group than in the PMSC group (Fig 11E). In summary, PMSC intervention can significantly change intestine dysmicrobiota of POI in which is mainly reflected in the significant increase of *Turicibacter* and *Desulfovibrio* and the decrease of *Alloprevotella*, *Parabacteroides*, *Rikenellaceae_RC9_gut_group* and *Rikenella*.

## PMSCs improved intestinal microbiota derived SCFAs in POI rats

Our research has uncovered significant findings about the role of PMSCs in improving SCFAs in the intestinal microbiota of POI rats. The linear regression equations and corresponding concentration ranges for each short-chain fatty acid standard are presented in Table 1. The critical metabolites of the intestinal microbiota, SCFAs mainly including acetic acid, propionic acid, butyric acid, isobutyric acid, isovaleric acid, valeric acid, and caproic acid, showed clear and easily identifiable peak shapes in the TIC chromatogram, which fully validates the accuracy of the analytical method and the reliability of the data (Fig 12A). In the RSD distribution map, the Y-axis represents the target metabolite's RSD value; the data quality is good when the RSD value is less than 15% (Fig 12B). Global metabolite clustering heat maps reveal metabolic patterns of metabolites under different experimental conditions. Metabolites with similar metabolic patterns tend to have similar functions or participate in the same metabolic processes or cellular pathways. The relative content of metabolites in the heat map is represented by the depth of color, with darker red indicating a higher expression level and darker blue indicating a lower expression level. The columns represent the different samples, while the rows indicate the names of the metabolites. The cluster tree on the left results from cluster analysis based on the similarity between metabolite molecules to show the correlation between metabolites and their grouping (Fig 12C). After detailed comparison, we found that the levels of acetic acid (P < 0.05), butyric acid (P < 0.05), and caproic acid (P < 0.01) in the MOD group were significantly reduced compared with those in the CON group (Fig 12E, 12F, 12I). However, other SCFAs show no significant differences (Fig 12D, 12G, 12H, 12J). In the PMSC group, compared with the MOD group, the level of caproic acid was significantly increased (P < 0.05, Fig 12I). In addition, compared with the CON group, the contents of butyric acid (P < 0.05), caproic acid (P < 0.05), and propionic acid (P < 0.05) in the MED group were significantly reduced (Fig 12F, 12I, 12J). In addition, KEGG (Kyoto Encyclopedia of Genes and Genomes)

**Table 1.  Linear regression equation and concentration range of each SCFAs standard.**

| Name | retention time (min) | Linear equation | correlation coefficient (r) | Linear range (ng/mL) | Limit of quantification (ng/mL) |
|---|---|---|---|---|---|
| Acetic acid | 4.33 | $y = 0.0251x + 0.0011$ | 0.9935 | 0.02-500 | 0.02 |
| Propionic acid | 5.36 | $y = 0.0289x + 9e\text{-}04$ | 0.9932 | 0.02-500 | 0.02 |
| Isobutyric acid | 5.74 | $y = 0.0355x + 5e\text{-}04$ | 0.993 | 0.02-500 | 0.02 |
| Butyric acid | 6.61 | $y = 0.0854x + 4e\text{-}04$ | 0.9944 | 0.02-250 | 0.02 |
| Isovaleric acid | 7.25 | $y = 0.0971x + 7e\text{-}04$ | 0.9932 | 0.02-250 | 0.02 |
| Valeric acid | 8.35 | $y = 0.0997x + 2e\text{-}04$ | 0.9953 | 0.02-250 | 0.02 |
| Caproic acid | 9.95 | $y = 0.0871x + 0.0017$ | 0.9953 | 0.02-250 | 0.02 |

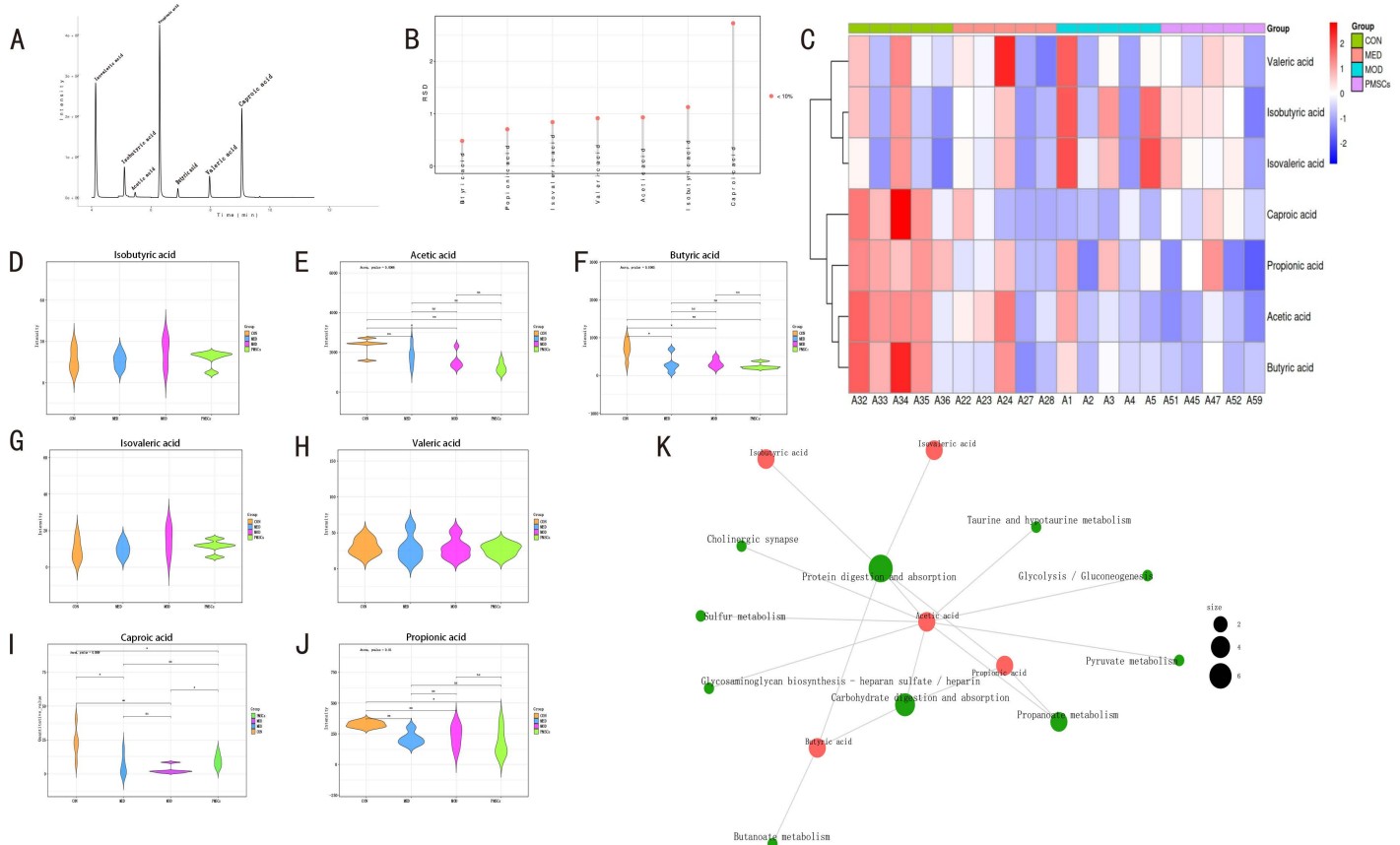

**Fig 12. Determination of SCFA content in fecal stool and intestinal metabolites of rats in each group.** A: Chromatogram of rat stool sample; B: RSD distribution diagram (Y-axis indicates the RSD value distribution of the target metabolite. RSD < 15%, indicating good value); C: Cluster heat map results; D-J: Contents of isobutyric acid, acetic acid, butyric acid, isovaleric acid, valeric acid, caproic acid and propionic acid (μg/g by gas chromatography-mass spectrometry) (n = 5/ group). The data were expressed as mean ± standard error (SEM) and were statistically analyzed using unpaired Student's t-test: * $p < 0.05$, ** $p < 0.01$.). K: KEGG metabolite molecular network diagram (Green dots represent metabolic pathways, other dots represent metabolite molecules (the size of the metabolic pathway site indicates the number of metabolite molecules associated with it; the larger the number, the dot). The molecular points of metabolites represent the magnitude of log2(FC) value through gradient change, and there is no metabolite log2(FC) information in many groups.

provided a strong support for metabolic analysis and metabolic network research. In Fig 12K, the green nodes represented metabolic pathways, while the other nodes displayed metabolite molecules. The size of the node of a metabolic pathway reflected the number of metabolite molecules associated with it, and the larger the number, the larger the node. The color change of the metabolite molecular nodes indicated the magnitude of the log2(FC) value, and the metabolites that did not label the log2(FC) information indicated that they did not differ significantly between the diverse groups. This analysis further revealed the critical role of SCFAs such as butyric acid in body metabolism, reinforcing the reliability of our data.

## Correlation analysis of intestinal flora with SCFAs, hormones, and inflammation in POI rats after PMSC treatment

We conducted a series of correlation analyses to analyze further the relationship between intestinal flora, hormones, inflammation, and SCFAs in POI (Fig 13). At the classification level, we observed a significant positive correlation between the abundance of *Firmicutes* and the content of caproic acid (P < 0.01, Fig 13A). In contrast, the abundance of *Bacteroides* was negatively

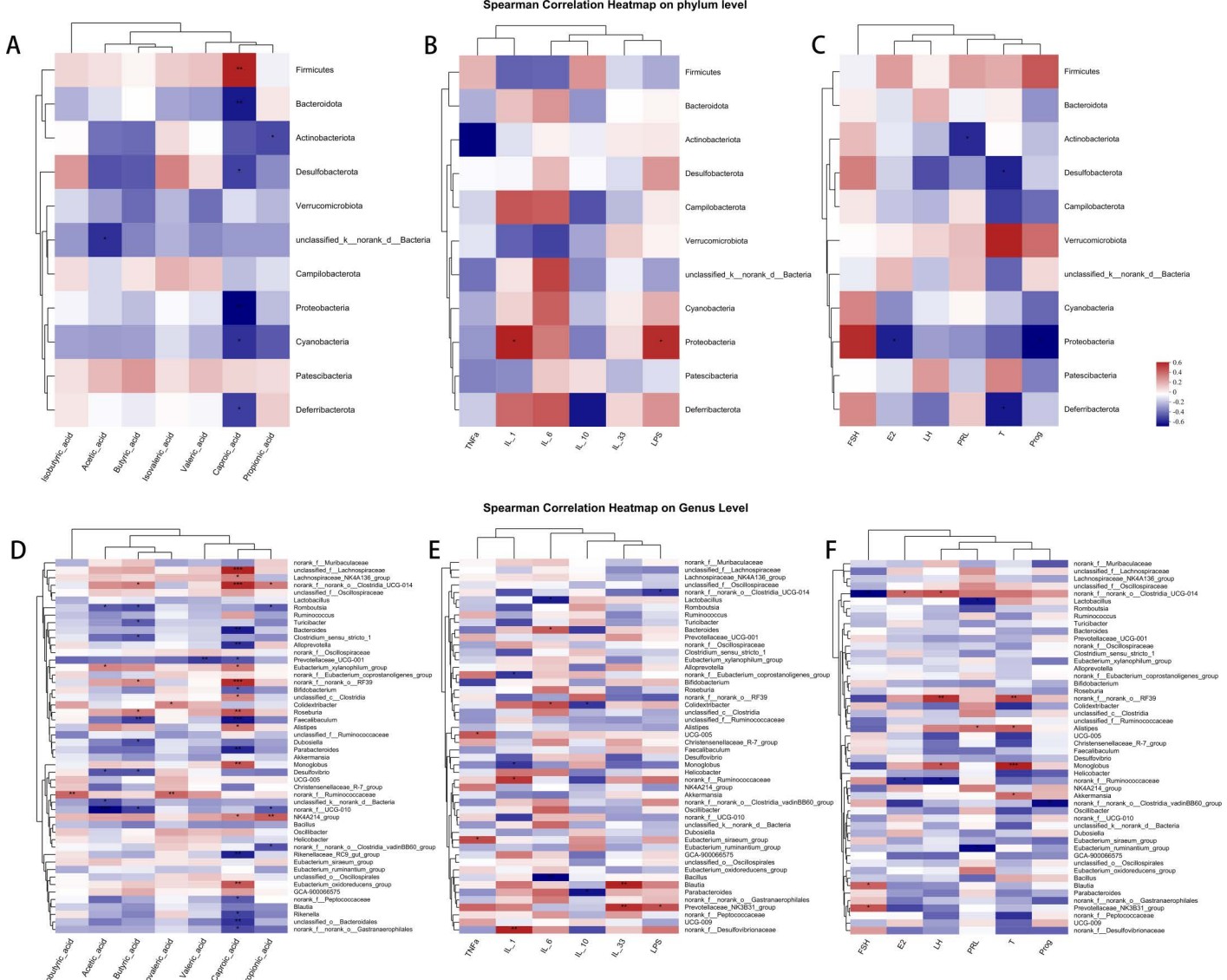

**Fig 13. Heat map of correlation between intestinal flora, SCFAs, hormones, and inflammation in POI rats after PMSC treatment.** A: Analysis of the correlation between bacteria and SCFAs at the gate level. B: Analysis of the correlation between bacteria and inflammatory factors at the gate level. C: Analysis of the correlation between bacteria and hormones at the gate level. D: Analysis of correlation between bacteria and SCFAs at the generic level. E: Analysis of correlation between bacteria and inflammatory factors at the generic level. F: Analysis of correlation between bacteria and hormones at the generic level. Spearman correlation analysis was used, and the intensity of correlation coefficients among samples was visually demonstrated by color depth. A total of 5 samples/groups were included in this study for analysis, and the results were marked by significance test (*$p < 0.05$, **$p < 0.01$, ***$p < 0.001$).

correlated with the content of caproic acid (P < 0.05, Fig 13A). These findings suggest a direct influence of the gut microbiome on SCFA production. On the other hand, the abundance of *Proteobacteria* was positively correlated with inflammation-related indicators such as LPS (P < 0.05) and IL-1β (P < 0.05, Fig 13B), while negatively correlated with steroid sex hormone E2 (P < 0.05, Fig 13C), suggesting that gut microbes may indirectly influence the course of POI by influencing inflammation levels and interacting with sex hormone levels. Further analysis at the taxonomic level of the genus, we found that the abundance of *Lachnospiraceae* ($p < 0.001$) and *Clostridia* (P < 0.001) were significantly positively correlated with caproic acid levels (Fig 13D).

In addition, the abundances of *Blautia* and *Prevotellaceae* were positively correlated with IL-33 (P < 0.05, Fig 13E) and FSH (P < 0.05, Fig 13F). These data further reinforce the vital link between gut microbes, inflammation, and POI-related hormones. We also found a significant correlation between butyric acid and various bacteria such as *Desulfovibrio*, *Dubosiella*, *Faecalibaculum*, *Roseburia*, *Turicibacter*, *Romboutsia*. These findings shed light on the complex network of interactions among gut flora, inflammation, steroid sex hormones, and metabolites SCFAs that collectively contribute to the disease progression of POI.

## Discussion

POI is a significant challenge in women's health, characterized by premature deterioration of ovarian function, which can lead to severe problems such as infertility and early menopause [2]. With the deepening of stem cell research, PMSCs are becoming a bright new star in treating POI due to their excellent regenerative potential [8]. PMSCs not only show great potential and promise in the treatment of ovarian dysfunction (including premature ovarian failure) [8]. Its unique immunomodulatory function also helps to alleviate the immune inflammatory response [23] within the ovaries, thereby protecting ovarian function. In addition, PMSCs are abundant in resources and easy to obtain and cultivate, providing an ideal cell source for treating POI [8]. We successfully isolated and purified PMSCs from fresh placentas, and these cells demonstrated excellent multifaceted differentiation potential, including the ability to differentiate into adipocytes, osteoblasts, and chondroblasts. In addition, through the analysis of cell markers, we confirmed that the mesenchymal markers CD105, CD73, and CD90 were highly expressed in PMSCs.

In contrast, the hematopoietic stem cell markers CD34, CD45, and monocyte marker CD14 were relatively low or non-expressed. Notably, these PMSCs do not express the central histocompatibility antigen II molecule HLA-DR, demonstrating their high purity and mesenchymal properties. These high-quality seed cells will provide a solid guarantee for subsequent research and promote the further exploration and application of PMSCs in the field of POI therapy.

To further explore the therapeutic effect of PMSCs on early-onset ovarian dysfunction (POI), we constructed a POI rat model [24]. By calculating and comparing the ovarian index of the model rat [25], we observed a significant decrease in ovarian weight in the MOD group (model group) compared to the CON group (control group) and PMSC group (placental mesenchymal stem cell treatment group). This finding validates the successful construction of the POI model and highlights the severe damage of POI to ovarian function. Notably, ovarian weight increased significantly in the PMSC group compared to the MOD group, suggesting that PMSCs positively affect ovarian function in POI rats. In addition, we also found a reduction in ovarian weight in the MED group compared to the CON group, which may be related to side effects of hormone therapy or insufficient hormone dosage. By further analyzing the ovarian index of rats, we found that the CON group and PMSC group were higher than the MOD group, which further confirmed the improvement effect of PMSCs on the ovarian function of POI rats. As the storehouse of germ cells and the center of female reproduction, the life expectancy of the ovaries is determined by the total number of germ cells [26]. The ovarian response can be evaluated by looking at changes in the number of mature oocytes [27]. The increase in the ovarian index in the PMSC group in this study may indicate the recovery of ovarian function and an increase in the number of follicles, thus offering hope for improving fertility in POI rats [28].

In this study, vaginal smears combined with Wright-Giemsa staining were used to observe the effects of PMSCs on the estrous cycle of POI rats [29]. The rats in the CON group showed

a standard 4-5 days estrous cycle, covering the pre-estrous period, the estrous period, the post-estrous period, and the inter-estrous period. This fully demonstrates the reproductive physiological state of normal rats [30]. In contrast, rats in the MOD group showed a persistent post-estrous period after seven days of intervention, which indicated that the ovarian function of POI rats had undergone significant changes and led to the disturbance of the estrous cycle. This finding is consistent with the typical changes in early-onset ovarian imfunction [31], demonstrating the model's validity. After the intervention of PMSCs or estrogen drugs, the estrous cycle of rats gradually returned to normal. This indicates that both PMSCs and estrogen can effectively improve the ovarian function of POI rats and promote the recovery of the estrous cycle. However, the estrous cycle of MOD group rats was constantly disturbed. It remained for a certain period during the whole experiment, indicating that the damage of POI to the reproductive function of rats was sustained, and it wasn't easy to recover spontaneously [32]. The pathological observation of ovarian tissue was made using hematoxylin and eosin (H&E) staining. The pathological changes of ovarian tissue in the MOD group confirmed that the cisplatin-induced POI model resulted in severe impairment of ovarian function in rats, which was manifested as follicular dysplasia and ovulation dysfunction [33]. After estrogen treatment, the number of primary follicles in the MOD group did not show significant changes. However, after PMSC intervention, the tissue structure of the ovaries was significantly improved. This result confirms the positive effect of PMSCs on ovarian function in POI rats and suggests a potential therapeutic impact in promoting follicle development [34] and preventing follicle atresia.

In this study, six plasma sex hormones were detected using an ELISA kit to comprehensively evaluate the effects of PMSCs on sex steroid hormones in early-onset POI model mice. The results showed that there were no significant differences in the levels of PROG and PRL among the groups, suggesting that these two hormones may not be the critical factors in the occurrence and development of POI or that the regulatory effect of PMSCs on them is not apparent under the conditions of this experiment. However, in terms of E2, we found that the MOD group had significantly lower E2 levels compared to the CON and MED groups, which is consistent with POI-induced decline in ovarian function [35]. Estrogen plays a vital role in many aspects of female reproduction, endocrine, and metabolism, and its decreased level may be one of the essential reasons for the decreased ovarian function in POI patients [36]. Although the MOD and PMSC groups did not achieve a statistically significant difference in the absolute value of E2 levels, we observed a possible underlying trend of difference between the two groups, suggesting that PMSCs may have restored estrogen levels to some extent in POI model mice. Elevated FSH levels in the MOD group may be due to feedback regulation due to ovarian hypofunction [37]. However, after PMSC intervention, FSH levels decreased significantly, suggesting that regulating FSH levels during the improvement of the POI model by PMSCs may be an essential mechanism, but further research is still needed. LH plays a vital role in a woman's menstrual cycle and reproductive function, and increased levels may be an essential factor in POI-induced ovarian dysfunction [38]. However, after PMSC intervention, FSH levels decreased significantly, suggesting that regulating FSH levels during the improvement of the POI model by PMSCs may be an essential mechanism, but further research is still needed. LH plays a vital role in a woman's menstrual cycle and reproductive function, and increased levels may be an essential factor in POI-induced ovarian dysfunction [39]. The testosterone T level in the MOD group is lower than in the MED group, suggesting that the POI model mice may have abnormal androgen synthesis or metabolism [40]. Taken together, PMSCs have a specific regulatory effect on steroid sex hormones in POI model mice, especially in E2 and FSH. However, its exact mechanism still needs to be further studied.

Inflammation plays a vital role in the occurrence and development of POI disease [41]. In this study, the levels of inflammation-related factors in the plasma and ovarian tissue of rats in each group were measured to explore whether PMSCs could ameliorate disease by inhibiting the inflammatory response. Firstly, we observed that plasma IL-1β level in the MOD group was significantly higher than in the CON, MED, and PMSC groups. This finding highlights the intense inflammatory response that rats in the MOD group may experience [42]. It is worth noting that the level of IL-1β in the ovarian tissue of the MOD group is also higher than that of the CON and MED groups, which further confirms the existence of local ovarian inflammation and suggests that ovarian inflammation may be one of the primary sources of IL-1β, which is consistent with the results of previous studies [43]. Secondly, the level of IL-6 in the ovarian tissue of POI rats was significantly increased, suggesting that the high expression of IL-6, a multifunctional proinflammatory cytokine, in the ovaries may further exacerbate the inflammatory damage of the ovarium [44]. At the same time, changes in plasma levels of IL-6 did not show statistical difference, which may mean that the local role of IL-6 is more important than its role in circulation. The level of TNF-α in MOD ovarian tissue was significantly higher than in CON and PMSC groups, further clarifying the serious ovarian inflammation in MOD group could be rectified by PMSCs treatment [45]. Plasma IL-33 level was also increased dramatically in the POI disease group, suggesting that IL-33 may be related to the progression of ovarian inflammation [46]. Its elevation may promote the inflammatory response of MOD group rats. In addition, the anti-inflammatory factor IL-10 was also significantly elevated in the PMSC group. These results suggest that in the process of improving POI disease by PMSC, the inflammatory response is inhibited dramatically, which is reflected in the above abnormally elevated proinflammatory factors (IL-1β, IL-6, TNF-α, IL-33) are significantly improved and the anti-inflammatory factor (IL-10) [47] is increased.

The abnormal increased LPS concentration in plasma of POI was downregulated after PMSC intervention, suggesting that PMSCs may enhance intestinal barrier function by affecting intestinal microbial balance and permeability. Studies have shown that ectopic LPS in the blood may be an essential factor in causing POI circulation and ovarian inflammation and can even be used to produce POI models [48]. Regarding mechanism, LPS binds to Toll-like receptor (TLR) -4 on the surface of inflammatory cells to promote the release of IL-6 and TNF-α through nuclear factor (NF-κB), as well as promote the release of inflammatory factor IL-1β through NLRP3 (Nucleotide-binding domain, leucine-rich - containing family, pyrin domain - containing-3) inflammasome pathway by mediating the inflammatory response. NF-κB can aggravate the inflammatory response by up-regulating the expression and formation of NLRP3 inflammasome molecules. In addition, IL-33 can enhance the LPS-mediated inflammatory response by binding to its receptor ST2 [49].

Previous studies have confirmed that POI has a significant imbalance of intestinal flora [50]. To verify the presence of gut microbiota dysbiosis in the POI rat model in this study and whether the improvement of PMSC was related to remodeling the intestinal flora, the composition was analyzed using 16S rRNA sequencing. β diversity results confirmed that POI rats' overall intestinal flora composition was significantly changed compared with the control group. Further analysis showed that at the phylum level, *Firmicutes* and *Bacteroidetes* accounted for about 90% of the intestinal bacteria in the four groups. In addition, *Actinobacteria* and *Desulfobacterota* account for about 8 percent of the total. At the same time, other species are relatively low in abundance, which is consistent with previous findings on the composition of gut microbes and provides further evidence for our understanding of gut microecology. The disturbance of intestinal flora in POI rats was mainly reflected in the dominant bacterium *Bacteroideta*, suggesting that it may play an important role in the occurrence and development of the disease, which is consistent with the results of a recent study on POI

[38]. Within the gastrointestinal microbiota, *Bacteroides* have a vast metabolic potential and are considered one of the most stable parts of the gastrointestinal microbiota. Studies have shown that metagenomic enrichment of *Bacteroides* is closely associated with obesity, insulin resistance, dyslipidemia, and inflammatory phenotypes [51,52]. In addition, the flora disturbance of POI rats at the phylum level was also reflected in the significantly increased levels of *Actinomyces*, *unclassified_k_norank_d_Bacteria*, *Proteobacteria*, *Cyanobacteria*, *Desulphuricides* and *Siderobacteria*. These changes may reflect the imbalance of intestinal microecology under POI. This is consistent with known POI-related microbial community findings [38]. In particular, increases in *Actinomyces desulphurizes*, and *hydrolytic bacteria* may be associated with inflammatory or metabolic changes caused by POI, and increases in this flora may further affect gut health. Interestingly, compared with the MOD group, the PMSC group showed significant differences in the relative abundance of major component bacteria, including *Firmicutes*, *Bacteroides*, and *Proteobacteria*, suggesting that PMSC treatment significantly impacted the level of intestinal microbial community structure. It is worth noting that *Firmicutes* and *Bacteroidetes* are the two most essential species in the human gut [53], and they play an important role in maintaining intestinal homeostasis, promoting nutrient absorption, and regulating immunity. Therefore, the effect of PMSC treatment on these species may be closely related to its therapeutic effect. However, it is worth noting that there was no significant difference in the abundance of *Firmicutes* and *Campilobacterota* between the MED and PMSC groups. This finding suggests that although the PMSC and MED groups differ in their treatment approaches, they may have similar effects in regulating specific gut microbiota.

When comparing the differences in the intestinal flora of rats in the PMSC and the MOD groups, we analyzed the intestinal flora of rats at the genus level. We found significant differences in the intestinal flora between different groups. We found that after PMSCs intervention, the relative abundance of *Unclassified_f_Lachnospiraceae*, *norank_f_norank_o_Clostridia_UCG-041* and *norank_f_norank_o_RF39* were significantly higher than those in MOD group. Most of these genera belong to *Firmicutes*. Studies have shown that it is crucial in maintaining intestinal homeostasis and promoting energy metabolism and nutrient absorption. In addition, there were also eight species of bacteria in the MOD group, such as *unclassified_k_norank_d_Bacteria* and *norank_f_UCG_010*, whose relative abundances were significantly higher than those in the CON group. However, the specific functions of these bacteria genera are still unclear, and further studies are needed to reveal them. In addition, PMSC intervention can significantly increase the abundance of *Turicibacter* and *Desulfovibrio*, while *Alloprevotella*, *Parabacteroides*, *Rikenellaceae_RC9_gut_group*, and *Rikenella* decreased substantially after the intervention. Recent studies have shown that succinic acid can activate immune cells through its specific surface receptor, succinic acid receptor 1 (SUCNR1), and enhance inflammation. However, the role of succinic acid in intestinal mucosal immune system inflammation remains unclear [54]. These findings suggest that PMSCs may inhibit inflammatory damage in POI rats by reducing succinic acid production by *Alloprevotella*. *Parabacteroides*, a gram-negative anaerobic bacterium, represents an essential component of the digestive tract microbiota. Butyric acid has been proved to serve as an essential metabolic product of anti-inflammatory *Enterobacteria*. The results suggest that PMSCs may inhibit POI inflammation by inhibiting *Rikenellaceae* to increase butyric acid synthesis. *Rikenella* also belongs to a group of anaerobic gram-negative bacteria in Bacteroidetes that are involved in inflammation by producing succinic acid [55], suggesting that PMSCs may play a role in improving POI by significantly affecting the bacteria and the succinic acid they produce. In addition, this study also found that the relative abundance of *Romboutsia*, *Turicibacter*, and *Dubosiella* in the MED group was significantly higher than that in the MOD group. The relative abundances of u*nclassified_k_norank_d_Bacteria*, *UCG_005*, and *Oscillibacter* were reduced considerably. These results suggest that the gut microbiota in the MOD group can be

improved by positive drug intervention and that specific microbiota regulation is different from PMSC intervention. The decrease in the relative abundance of *Ruminococcus* in the MED group may reflect the reduction in the cellulose-degrading ability. This may impact the host's energy metabolism and nutrient absorption. Notably, the relative abundances of *norank_f_UCG_010*, *UCG_005*, and *Christensenellaceae_R-7_group* in the MED group were also lower than that in the PMSC group. The reduction of these microbial communities may imply a disruption of the gut microecological balance, which may affect host immune function and metabolic processes, thereby affecting the occurrence and development of POI.

This finding is significant in understanding the effects of PMSCs on POI rats and the role of SCFAs in promoting health and facilitating a variety of diseases [56]. However, the role and mechanism of SCFAs have not been reported for POI diseases. Therefore, this study focused on the effect of PMSCs on SCFAs produced by intestinal microbiota in POI rats. As a critical metabolite of intestinal microbiota, SCFAs are essential in maintaining intestinal health, promoting energy metabolism, and regulating the immune system [57]. We found that compared with the CON group, acetic acid, butyric acid, and caproic acid levels in the gut of MOD group rats were significantly reduced, suggesting a significant deficiency of SCFAs in POI, which may be involved in the occurrence and development of the disease. Acetic acid and butyric acid are the two main SCFAs essential for maintaining the integrity of the intestinal barrier and promoting energy metabolism in intestinal cells [38]. However, it is significantly deficient in POI rats, suggesting that the rectification of acetic acid and butyric acid deficiency can improve the disease. In addition, studies have shown that caproic acid inhibits inflammatory cytokines such as IL-8, IL-6, and TNF-α by inhibiting MAPK phosphorylation and NF-kB activation. However, in terms of inhibiting inflammation, more studies, including our research group, point to butyric acid acting as a crucial role in controlling inflammation, which is mainly achieved by G-protein-coupled receptors (GPCR) and histone deacetylase (HDAC) inhibitors [58,59]. In addition, the levels of acetic acid, butyric acid, caproic acid, and propionic acid in the MED group were also significantly different from those in the CON group, and the levels of butyric acid, caproic acid, and propionic acid were decreased. This finding further highlights the potential of SCFAs to improve the health of POI rats. However, we also noted that while some SCFA levels in the gut of the PMSC group were improved, they still did not fully return to the levels of the CON group. This could mean that the effects of POI on the gut microbiome are complex and long-lasting, requiring more extended treatment or more sophisticated intervention strategies to reverse entirely.

Our studies have confirmed that POI is a multi-factorial disease, but the exact mechanism is unclear for most patients, suggesting a complex multi-factorial interaction [2]. This study, a collaborative effort, investigated the relationship between intestinal flora and SCFAs, hormones, and inflammation in POI rats after PMSC treatment. Our collective efforts have led to a deeper understanding of the disease. We revealed the changes in the intestinal microbial community during the pathological process of POI and its interaction with the host metabolism and immune system. Through correlation analysis, we found that the abundance of *Firmicutes* was significantly positively correlated with the level of caproic acid at the phylum level, which supported the critical role of Firmicutes in generating SCFAs. *Firmicutes*, a dominant bacterial group in the intestine, and SCFAs produced by their metabolism, play an indispensable role in maintaining intestinal homeostasis and promoting body health. The results of this study further reveal the vital link between gut microbes and SCFAs and show that an increase in firmicutes contributes to the production of caproic acid. Notably, we found a significant negative correlation between the abundance of *Bacteroides* and the level of caproic acid, indicating that *Bacteroides* may be related to the decrease of caproic acid in the gut of

POI rats. *Bacteroides* are another vital member of the intestinal microbiota, and the change in their abundance may reflect the intestinal microbiota imbalance and the pathological process of POI. This collaborative effort has brought us closer to understanding the disease and developing effective treatments. In addition to our significant findings, we also uncovered potential implications that could lead to new treatments and interventions. We found that *Proteobacteria* was positively correlated with LPS and IL-1β while negatively correlated with estrogen E2. *Proteobacteria* is a species closely related to inflammation and infection, and its increased abundance may indicate intestinal inflammation. At the same time, LPS and IL-1β are essential markers of the inflammatory process, and their elevated levels indicate an inflammatory response in the gut of POI rats. However, the negative correlation between *Proteobacteria* and E2 suggests that the decrease in estrogen level caused by POI may be related to the changes in intestinal microflora. Through statistical analysis, we found that the abundance of *Trichospiridae* and *Clostridium* were significantly positively correlated with caproic acid levels. This finding further confirmed the close relationship between gut microbiota and SCFAs. *Trichospiridae* and *Clostridium* are essential members of the intestinal microbial community, and their abundance changes may directly affect the synthesis and secretion of caproic acid. As a critical SCFA, caproic acid has many functions, such as regulating intestinal pH and promoting intestinal health. Therefore, an increase in these flora may contribute to increasing levels of caproic acid, thereby maintaining intestinal homeostasis and host health. In addition, we found that the abundance of *Blautia* and *Prevotellaceae* was positively correlated with IL-33 and FSH levels. Increased levels of IL-33, an important pro-inflammatory factor, may reflect increased intestinal inflammation [60]. At the same time, FSH, an essential reproductive hormone, plays a vital role in female reproductive health [61]. In this study, we found that the abundance of *Blautia* and *Prevotellaceae* was positively correlated with FSH levels, suggesting that changes in gut microbiota may be closely related to the function of the reproductive system. Moreover, we found that the contents of various butyric-producing bacteria and butyric acid were reduced, suggesting that supplementing butyric acid and its producing bacteria may positively affect the prevention and treatment of POI disease. Collectively, there is a complex and close correlation and interaction among PMSCs intervention, intestinal flora, SCFAs, steroid sex hormones, and inflammatory indicators, jointly contributing to the occurrence and development of diseases. Our findings open up new possibilities for the prevention and treatment of POI.

Thus, this study preliminarily revealed the effects of PMSC transplantation on POI rats and the mechanisms related to sex steroid hormones, inflammation, SCFAs, and intestinal microbiota. These findings help us understand the pathogenesis of POI more comprehensively and provide a theoretical basis for the clinical application of PMSCs in treating this disease. However, this study still has some limitations, such as a relatively small sample size and a short experiment period. Overcoming these limitations is crucial for expanding the research scale further and extending the experimental period, which will ultimately lead to more robust and applicable findings.

## Conclusion

PMSC transplantation holds immense potential in the treatment of POI. By regulating steroid sex hormone secretion, inhibiting inflammation, and affecting gut microbiota and SCFA metabolism, PMSCs could be a game-changer in the field of reproductive health. This study's findings not only deepen our understanding of POI's pathogenesis but also reveal the promising mechanism of PMSCs in its treatment. This provides a strong theoretical basis for future clinical applications and encourages us to explore the full potential of PMSC transplantation in treating POI.

## Supporting information

**S1 File. Datasets of each histogram.**
(DOCX)

## Author contributions

**Conceptualization:** Hao Wang, Xiaoxia Zhang.

**Data curation:** Shudan Liu.

**Formal analysis:** Ting Wang.

**Funding acquisition:** Qikuan Hu, Hao Wang.

**Investigation:** Yiwei Li, Junbai Ma.

**Methodology:** Yuanyuan Liu.

**Project administration:** Hao Wang, Xiaoxia Zhang.

**Software:** Ting Wang, Yiwei Li.

**Supervision:** Qikuan Hu.

**Validation:** Yuanyuan Liu, Junbai Ma.

**Writing – original draft:** Shudan Liu.

**Writing – review & editing:** Hao Wang.

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
