## [Decision Letter · Decision Letter 0]

27 Sep 2024

PONE-D-24-36829Human placental mesenchymal stem cells ameliorates premature ovarian insufficiency via modulating gut microbiota and suppressing the inflammation in ratsPLOS ONE

Dear Dr. Wang,

Thank you for submitting your manuscript to PLOS ONE. After careful consideration, we feel that it has merit but does not fully meet PLOS ONE’s publication criteria as it currently stands. Therefore, we invite you to submit a revised version of the manuscript that addresses the points raised during the review process.

We look forward to receiving your revised manuscript.

Kind regards,

Jiajie She

Academic Editor

PLOS ONE

Reviewers' comments:

Reviewer's Responses to Questions

**Comments to the Author**

1. Is the manuscript technically sound, and do the data support the conclusions?

Reviewer #1: Partly

Reviewer #2: No

Reviewer #3: Partly

Reviewer #4: Partly

2. Has the statistical analysis been performed appropriately and rigorously? 

Reviewer #1: Yes

Reviewer #2: No

Reviewer #3: Yes

Reviewer #4: Yes

3. Have the authors made all data underlying the findings in their manuscript fully available?

Reviewer #1: Yes

Reviewer #2: Yes

Reviewer #3: Yes

Reviewer #4: No

4. Is the manuscript presented in an intelligible fashion and written in standard English?

Reviewer #1: No

Reviewer #2: No

Reviewer #3: Yes

Reviewer #4: No

5. Review Comments to the Author

Reviewer #1: The action mechanism of placental mesenchymal stem cells on premature ovarian insufficiency induced by cisplatin was discussed. There are the following problems to be solved:

1. When first using an abbreviation, provide the full term, such as in the Materials and Methods section when first mentioning SD rats, it should be noted as SD (Sprague Dawley).

2. Figures 2, 5, and 6 contain pathological images, which should include a scale bar on the images. The specific information of the scale bar should also be indicated in the Figure legends section. Additionally, the images, particularly those in the gut microbiota section, are not clear enough, which affects readability.

3. In the discussion section, "In addition, the flora disturbance of POI rats at the phylum level was also reflected in the significantly increased levels of actinomyces ...". The initial letter of “actinomyces” should be capitalized, please check the rest of the article to ensure adherence to spelling standards.

4.In the discussion section, "the PMSC group showed significant differences in the abundance of major component bacteria," should be "relative abundance."

5.In the discussion section, the changes in the gut microbiota do not adequately explain their role in improving premature ovarian insufficiency. For example, "Lachnospiraceae, which mainly exists in the intestinal flora of humans and mammals, belongs to the anaerobic bacteria, which can prevent the occurrence of colorectal cancer by producing butyric acid," "Ruminococcus, a vital cellulose-degrading bacterium, plays an irreplaceable role in maintaining intestinal microecological balance and promoting the overall health of the host." It is suggested that the authors focus their discussion on premature ovarian insufficiency rather than expending excessive narrative on the beneficial effects of the gut microbiota and its metabolic products on the host.

6.In this study investigating premature ovarian insufficiency, why were fecal samples collected to examine changes in the gut microbiota instead of scraping vaginal secretions to explore changes in the vaginal flora?

7.Compared to the targeted short-chain fatty acid detection used in this article, non-targeted metabolomics detection might provide more meaningful results.

8.If conditions permit, it is recommended to supplement a proteomic analysis of the ovaries, which could better infer the potential mechanisms of PMSC treatment for premature ovarian insufficiency. Moreover, the results of the proteomic analysis could be jointly analyzed with the 16S rDNA sequencing results.

9.premature ovarian insufficiency primarily affects fertility. If possible in the future, it could be considered to observe the effects of PMSC on the fertility of a premature ovarian insufficiency rat model, as well as the impact on the offspring of the premature ovarian insufficiency rat model.

Reviewer #2: Effectiveness of PMSCs on Ovarian Function: The results indicate that PMSC intervention improved ovarian weight and index in POI rats. How do these improvements compare quantitatively with other potential treatments for POI, such as hormone replacement therapy or other stem cell-based therapies?

Estrous Cycle Restoration: The study mentions that PMSCs helped restore the estrous cycle in POI rats. What mechanisms are proposed for PMSCs influencing the regularization of the estrous cycle, and were there any observed side effects during this restoration process?

Inflammatory Markers: The results show that PMSCs reduced the levels of pro-inflammatory cytokines such as IL-1β and IL-6 in plasma and ovarian tissue. Were these reductions sustained over time, and how do they correlate with long-term improvements in ovarian function?

Microbiota Changes: The study reports significant changes in gut microbiota composition following PMSC intervention. Which specific bacterial changes are hypothesized to be directly linked to ovarian recovery, and are these changes expected to persist long-term without continuous treatment?

SCFA Levels: Short-chain fatty acids (SCFAs) like caproic acid were significantly altered by PMSC intervention. How are these changes in SCFA levels mechanistically linked to improved ovarian function, and could SCFAs be potential biomarkers for therapeutic success?

LPS Levels and Endotoxemia: The reduction in plasma LPS levels suggests improved intestinal barrier function. Is there any evidence to support a direct relationship between the reduction in LPS levels and improvements in ovarian histopathology, or is this effect secondary?

Comparison with Control Groups: The results show that PMSC treatment had a more significant impact on ovarian function and inflammatory markers than hormone therapy (MED group). What specific aspects of PMSC treatment are believed to account for its superior efficacy compared to conventional hormone therapy?

Hormonal Regulation: While E2 levels were improved by PMSC intervention, they did not reach statistical significance compared to the control. What might explain the relatively modest impact on estrogen levels, and could different dosing or timing of PMSC administration enhance this effect?

Long-Term Efficacy: Has the study assessed the long-term effects of PMSCs on ovarian function, inflammatory markers, and gut microbiota, particularly after treatment cessation? How durable are the therapeutic benefits observed in this study?

Reviewer #3: Thanks for your exciting piece of work. I have listed some general and technical issues. However,

Due to the low resolution of figures 9-13 and the lack of line numbers, I couldn’t properly review the manuscript.

General Issue,

1- Define all abbreviations in the abstract

2- Show the positive or negative correlation of the mentioned phyla in the abstract and not just say: “remarkable difference.”

3- Mention the concentration of collagenase A and DNase for placenta cell isolation

4- Why didn’t you use the shaker incubator for digestion?

5- Mention the yield of cell isolation and the reference of the method you used

6- Figures 9-13 are of low quality, and I am not able to review the section of this manuscript that comprises them.

Technical issues

1- How come you used one fluorochrome (PE) for different markers (CD45, CD73, CD90, and CD105) and FITC for HLA-DR, CD34, and CD14? So, the flow cytometry analysis (Fig. 2E) is not understandable on the first page of your results.

2- I can not understand the meaning of Control cells in this sentence (At the same time, control cells continued to be added to the mesenchymal stem cells-specific culture medium.)

3- In the result section entitled:” MSCs intervention ameliorated the homeostasis of steroid hormones in rats with POI,” it seems that in the PMCS group, the level of LH is significantly higher than in MOD, which is not reflected in the graph. Can you check?

4- According to Figure 8, the LPS graph, all groups have the same LPS. Please change the information in the results accordingly

5- Please include the effect of PMCS regarding the microbial changes among these lines at page 22:

“Analysis results showed that the MOD group was enriched in

Rikenella, Micrococcaceae, Bacteroidales, and Desulfovibrionaceae, while the CON group enriched in Lachnospiraceae and Clostridia (Fig. 9E). These findings provided important clues for us to understand further the effects of PMSCs on POI by modulating the intestinal microbiota”.

6- The information you provided for this title:” The difference in intestinal flora in rats of each group at the phylum level,” has been repeated:” The difference in intestinal flora in rats of each group at the genus level.” You may need to omit from the second part.

Final Note: Despite having trouble interpreting the results for Figures 9-13, I believe that the discussion of gut microbiota, inflammatory cytokines, and SCFA was comprehensive and informative. Therefore, I asked the corresponding editor to send me the high-resolution figures and a version of the manuscript with the line number to finish my review.

Reviewer #4: This peer-reviewed manuscript is devoted to the current topic of research on gut microbiota composition in association with the different health status of the host. This study aimed to investigate the effects and potential mechanism of human placental mesenchymal stem cells (PMSCs) on improving early-onset ovarian dysfunction (POI) in a rat model.

Unfortunately, the authors did not use line numbers in the document; thus, it was difficult to specify the comments.

1. The first comment concerns terminology. I recommend replacing "intestinal" with "gut", since the work did not study the parietal microbiota, but the microbiota that passes through the tract and comes out in the form of feces.

2. The methods section "Intestinal microbiota analysis" needs to be further assessed. For example, I doubt whether PCR was actually performed in "15 mL of Phusion® hi-Fi PCR Master Mix", and what was the final volume of the reaction mixture so that the concentrations of other components could be adequately recalculated? What sequencing mode was used by the authors?

3. The next comment is also terminological and structural. Correct: Firmicutes, Bacteroides, Proteobacteria, and so on in the text. There are no such taxa in modern bacterial taxonomy. The authors describe microbial communities; therefore, they must use modern bacterial taxonomy. Therefore, the entire taxonomy of the article must be as follows: https://www.bacterio.net/ and Oren A, Garrity GM. Valid publication of the names of forty-two phyla of prokaryotes. Int J Syst Evol Microbiol 2021; 71:5056.

In the text of the article, the authors use three variants of names for some taxa. In the sections "PMSCs intervention modulated the difference of overall community structure of intestinal microbiota", "The difference in intestinal flora in rats of each group at the phylum level", and "The difference of intestinal flora in rats of each group at genus level", the names of all bacterial taxa must be strictly consistent with the version presented in the corresponding figures. The authors did not describe the community at the species level; therefore, it is necessary to either provide an additional description or exclude this mention from the text.

4. I cannot assess the quality and consistency of the results in Figure 12 because it is of very low resolution; in Fig. 12C, undeciphered marking appears. The drawing requires correction.

Overall, the article requires additional correction and technical revisions.

6. PLOS authors have the option to publish the peer review history of their article (what does this mean? ). If published, this will include your full peer review and any attached files.

**Do you want your identity to be public for this peer review?** For information about this choice, including consent withdrawal, please see our Privacy Policy .

Reviewer #1: **Yes: ** Zhitong Deng

Reviewer #2: No

Reviewer #3: **Yes: ** Fariba Ghiamati Yazdi

Reviewer #4: No

---

## [Author Response · Author response to Decision Letter 1]

28 Oct 2024

Dear editor:

Thank you for your kind letter. Thanks for reviewers’ valuable comments on our manuscript. We have completely revised the manuscript in accordance with the comments, and carefully proof-read the manuscript to minimize typographical, grammatical and bibliographical errors. All errors have been modified and marked in red throughout the manuscript.

Reviewer #1: The action mechanism of placental mesenchymal stem cells on premature ovarian insufficiency induced by cisplatin was discussed. There are the following problems to be solved:

1. When first using an abbreviation, provide the full term, such as in the Materials and Methods section when first mentioning SD rats, it should be noted as SD (Sprague Dawley).

Reply: Thank you for your very kind advice. According to your suggestion, we have given the complete terminology of SD rats in the Materials and Methods section. And the whole manuscript was examined and supplemented as necessary.

2. Figures 2, 5, and 6 contain pathological images, which should include a scale bar on the images. The specific information of the scale bar should also be indicated in the Figure legends section. Additionally, the images, particularly those in the gut microbiota section, are not clear enough, which affects readability.

Reply: We really appreciate for your constructive suggestions. According to your advices, we have added appropriate scale bars to Figure 2, 5, and 6, and detailed the specific information of the scale bars in the legend section so that readers can better understand the actual size of the images. Moreover, we reprocessed the images of the intestinal microbiota section, to improve their resolution and clarity, ensuring that the images are clearer and easier to read.

3. In the discussion section, "In addition, the flora disturbance of POI rats at the phylum level was also reflected in the significantly increased levels of actinomyces ...". The initial letter of “actinomyces” should be capitalized, please check the rest of the article to ensure adherence to spelling standards.

Reply: Thank you for your very careful advice. According to your suggestion, we have capitalized the first letter of "Actinobacteria" and reviewed the rest of the manuscript to ensure adherence to spelling standards.

4.In the discussion section, "the PMSC group showed significant differences in the abundance of major component bacteria," should be "relative abundance."

Reply: We deeply appreciate for your professional advice. Based on your suggestion, we have adjusted this sentence to “the PMSC group showed significant differences in the relative abundance of major component bacteria,”.

5.In the discussion section, the changes in the gut microbiota do not adequately explain their role in improving premature ovarian insufficiency. For example, "Lachnospiraceae, which mainly exists in the intestinal flora of humans and mammals, belongs to the anaerobic bacteria, which can prevent the occurrence of colorectal cancer by producing butyric acid," "Ruminococcus, a vital cellulose-degrading bacterium, plays an irreplaceable role in maintaining intestinal microecological balance and promoting the overall health of the host." It is suggested that the authors focus their discussion on premature ovarian insufficiency rather than expending excessive narrative on the beneficial effects of the gut microbiota and its metabolic products on the host.

Reply: Thank you for your very rigorous suggestions. According to your advice, we have revised the discussion section, reducing the detailed description of the gut microbiota (such as Lachnospiraceae and Ruminococcus) and their metabolic products on the overall health of the host, and instead focusing more on the potential mechanisms and effects of these microbiota changes on POI. We have conducted a more in-depth discussion on their possible regulatory roles to avoid overextending into other areas.

6.In this study investigating premature ovarian insufficiency, why were fecal samples collected to examine changes in the gut microbiota instead of scraping vaginal secretions to explore changes in the vaginal flora?

Reply: We deeply appreciate for your insightful comments. Numerous studies have shown that the gut microbiota influences the host's endocrine system, immune system, and overall health through its metabolic products, thereby playing a regulatory role in various diseases. In the case of premature ovarian insufficiency (POI), the gut microbiota is believed to affect ovarian function by regulating hormone levels, metabolic pathways, and inflammatory states. Therefore, we chose to study changes in the gut microbiota to explore its potential role in POI. Additionally, fecal samples are relatively easy to collect and non-invasive, while also providing a comprehensive reflection of the composition and functional state of the gut microbiota. Thus, using fecal samples as the research subject is more appropriate for this study. As your valuable suggestion, vaginal microbiota in POI after the treatment will be further investigated.

7.Compared to the targeted short-chain fatty acid detection used in this article, non-targeted metabolomics detection might provide more meaningful results.

Reply: Thank you for your valuable suggestions. We understand that untargeted metabolomic testing can provide more comprehensive metabolite information and help reveal more underlying metabolic mechanisms. However, we chose targeted short-chain fatty acid (SCFA) testing in this study for several reasons: Firstly, the main objective of this study is to explore the role of specific metabolites of the gut microbiota, such as SCFAs, in POI. SCFAs are key metabolites of the gut microbiota, and their role in regulating inflammation, metabolism, and endocrine function has been widely studied. Therefore, targeted detection of SCFAs is more directly aligned with the research question of our study. Secondly, targeted detection provides higher sensitivity and accuracy in quantifying specific metabolites. In this study, quantifying SCFAs is crucial, as we aim to further reveal their potential mechanisms in POI through quantitative analysis.

We are very grateful for your suggestion, and we will consider incorporating non-targeted metabolomics analysis in future research to obtain more comprehensive metabolite information.

8.If conditions permit, it is recommended to supplement a proteomic analysis of the ovaries, which could better infer the potential mechanisms of PMSC treatment for premature ovarian insufficiency. Moreover, the results of the proteomic analysis could be jointly analyzed with the 16S rDNA sequencing results.

Reply: Thank you for your constructive suggestion. We agree that proteomic analysis can provide a more comprehensive view of protein expression and regulation in the ovaries, which is crucial for revealing the molecular mechanisms of PMSC treatment for POI. However, due to current experimental conditions and resource limitations, we are unable to conduct proteomic analysis at this time. Nevertheless, we will consider this suggestion as an important direction for future research and further refine our experimental design to explore the potential mechanisms of PMSC treatment.

9.Premature ovarian insufficiency primarily affects fertility. If possible in the future, it could be considered to observe the effects of PMSC on the fertility of a premature ovarian insufficiency rat model, as well as the impact on the offspring of the premature ovarian insufficiency rat model.

Reply: Thank you for your valuable suggestion. We completely agree that premature ovarian insufficiency (POI) primarily affects fertility, so observing the impact of PMSC (placental mesenchymal stem cells) on the fertility of a POI rat model, as well as the effects on the offspring, will provide a more comprehensive perspective for our research. We will incorporate your suggested research direction into our future plans. By examining the therapeutic effects of PMSC in a POI rat model, we can better assess its impact on fertility and explore the underlying mechanisms. Additionally, we will consider including fertility and offspring health as evaluation criteria to comprehensively assess the effectiveness of PMSC treatment and its effects on both the mother and offspring.

Reviewer #2:

1.Effectiveness of PMSCs on Ovarian Function: The results indicate that PMSC intervention improved ovarian weight and index in POI rats. How do these improvements compare quantitatively with other potential treatments for POI, such as hormone replacement therapy or other stem cell-based therapies?

Reply: We deeply appreciate for your significative advice. In this study, we primarily focused on the impact of PMSC on the POI rat model and demonstrated its effects on improving ovarian weight and index. However, due to the design limitations of this study, we have not conducted a direct comparison with other treatment methods. We fully agree with your suggestion and will consider conducting comparative studies between PMSC and hormone replacement therapy or other stem cell therapies in future research. Through such comparisons, we will be able to quantitatively assess the effectiveness of different treatment methods and provide a more comprehensive basis for the treatment of POI.

2.Estrous Cycle Restoration: The study mentions that PMSCs helped restore the estrous cycle in POI rats. What mechanisms are proposed for PMSCs influencing the regularization of the estrous cycle, and were there any observed side effects during this restoration process?

Reply: Thank you for your valuable question. PMSC may influence the regularization of the estrous cycle through the following mechanisms: 1)Hormonal Regulation: PMSC may promote ovarian functional recovery by secreting beneficial cytokines and growth factors, which in turn affect hormone levels and help restore the normal estrous cycle. 2)Anti-inflammatory Effects: PMSC possesses anti-inflammatory properties and may improve the ovarian microenvironment by alleviating the inflammatory response in the ovaries and surrounding tissues, thereby promoting ovarian functional recovery. 3)Stem Cell Differentiation: PMSC may differentiate into ovarian-related cells, directly participating in the restoration of ovarian function.

In this study, we closely monitored the overall health status and physiological changes of POI rats during PMSC intervention. So far, no significant side effects have been observed. However, due to the limitations in the duration and scope of observation in this study, we recommend that future research continue to monitor the long-term effects and potential side effects.

3.Inflammatory Markers: The results show that PMSCs reduced the levels of pro-inflammatory cytokines such as IL-1β and IL-6 in plasma and ovarian tissue. Were these reductions sustained over time, and how do they correlate with long-term improvements in ovarian function?

Reply: Thank you for raising this important question. While our study observed a significant reduction in pro-inflammatory cytokine levels following intervention, due to the limited time frame of this study, we have not yet been able to assess whether these reductions are sustained over a longer period. Future research will extend the observation period to determine the long-term anti-inflammatory effects of PMSC treatment. We hypothesize that the reduction in pro-inflammatory cytokine levels may be correlated with long-term improvements in ovarian function. A sustained anti-inflammatory state could help maintain a stable ovarian microenvironment, thereby supporting ovarian function recovery. However, further studies are needed to clarify this correlation and to evaluate the long-term therapeutic effects.

4.Microbiota Changes: The study reports significant changes in gut microbiota composition following PMSC intervention. Which specific bacterial changes are hypothesized to be directly linked to ovarian recovery, and are these changes expected to persist long-term without continuous treatment?

Reply: Thank you for your important question. We hypothesize that certain key changes in the gut microbiota are closely related to ovarian recovery. For example, bacterial groups such as Lachnospiraceae and Ruminococcaceae, which produce SCFA, showed significant increases after PMSC intervention. SCFAs are believed to have anti-inflammatory, metabolic, and endocrine regulatory effects, which may improve overall health and thereby promote ovarian function recovery. Currently, there is not enough evidence to determine whether these bacterial changes can be sustained long-term without continuous treatment. We speculate that some of the changes in gut microbiota might persist, particularly those closely associated with host health. However, given that the gut microbiota is susceptible to external factors such as diet, environment, and lifestyle, the long-term stability of these changes requires further research.

5.SCFA Levels: Short-chain fatty acids (SCFAs) like caproic acid were significantly altered by PMSC intervention. How are these changes in SCFA levels mechanistically linked to improved ovarian function, and could SCFAs be potential biomarkers for therapeutic success?

Reply: We deeply appreciate for your question. SCFAs have various biological activities, including anti-inflammatory, metabolic regulation, and immune modulation functions. We hypothesize that SCFAs may promote ovarian function improvement through mechanisms such as anti-inflammation, metabolic regulation, and the gut-ovary axis. Changes in SCFA levels can reflect improvements in gut microbiota and their metabolic functions, which may be closely related to ovarian function recovery. However, while SCFAs have the potential to become biomarkers, further clinical and mechanistic studies are needed to validate this hypothesis.

6.LPS Levels and Endotoxemia: The reduction in plasma LPS levels suggests improved intestinal barrier function. Is there any evidence to support a direct relationship between the reduction in LPS levels and improvements in ovarian histopathology, or is this effect secondary?

Reply: Thank you for your very kind advice. Currently, there is no direct evidence supporting a direct relationship between the reduction in LPS levels and improvements in ovarian histopathology. However, we hypothesize that the decrease in LPS levels may indirectly promote improvements in ovarian histopathology by reducing pro-inflammatory responses, thereby improving both systemic and local ovarian inflammation. We fully recognize the importance of further investigating the potential direct relationship between these factors and plan to explore the mechanistic links between changes in LPS levels and ovarian recovery in future studies.

7.Comparison with Control Groups: The results show that PMSC treatment had a more significant impact on ovarian function and inflammatory markers than hormone therapy (MED group). What specific aspects of PMSC treatment are believed to account for its superior efficacy compared to conventional hormone therapy?

Reply: Thank you for your valuable question. Regarding the superior impact of PMSC treatment on ovarian function and inflammatory markers compared to hormone therapy, we believe the advantages of PMSC stem from its diverse biological properties. First, PMSCs can secrete various bioactive molecules (such as growth factors, anti-inflammatory factors, and cytokines) and reduce both local and systemic inflammation through paracrine effects. In contrast, hormone therapy primarily focuses on endocrine regulation and does not fully address the inflammatory state. Additionally, PMSCs can modulate the immune system, balancing pro-inflammatory and anti-inflammatory responses, which helps maintain a healthy ovarian microenvironment. Hormone therapy, on the other hand, mainly works by supplementing exogenous hormones to balance hormone levels, with limited regulation of the immune system.

8.Hormonal Regulation: While E2 levels were improved by PMSC intervention, they did not reach statistical significance compared to the control. What might explain the relatively modest impact on estrogen levels, and could different dosing or timing of PMSC administration enhance this effect?

Reply: Th

---

## [Editor Report · Decision Letter 1]

31 Oct 2024

Human placental mesenchymal stem cells ameliorates premature ovarian insufficiency via modulating gut microbiota and suppressing the inflammation in rats

PONE-D-24-36829R1

Dear Dr. Wang,

We’re pleased to inform you that your manuscript has been judged scientifically suitable for publication and will be formally accepted for publication once it meets all outstanding technical requirements.

Kind regards,

Jiajie She

Academic Editor

PLOS ONE
---

## [Editor Report · Acceptance letter]

PONE-D-24-36829R1

PLOS ONE

Dear Dr. Wang,

I'm pleased to inform you that your manuscript has been deemed suitable for publication in PLOS ONE. Congratulations! Your manuscript is now being handed over to our production team.

Kind regards,

on behalf of

Dr. Jiajie She

Academic Editor

PLOS ONE